# Stability and Generalization in Free Adversarial Training

**Xiwei Cheng**                                                     *xwcheng@link.cuhk.edu.hk*
*The Chinese University of Hong Kong*

**Kexin Fu**                                                              *fu448@purdue.edu*
*Purdue University*

**Farzan Farnia**                                                   *farnia@cse.cuhk.edu.hk*
*The Chinese University of Hong Kong*

**Reviewed on OpenReview:** *https://openreview.net/forum?id=jmwEiC9bq2*

## Abstract

While adversarial training methods have significantly improved the robustness of deep neural networks against norm-bounded adversarial perturbations, the generalization gap between their performance on training and test data is considerably greater than that of standard empirical risk minimization. Recent studies have aimed to connect the generalization properties of adversarially trained classifiers to the min-max optimization algorithm used in their training. In this work, we analyze the interconnections between generalization and optimization in adversarial training using the algorithmic stability framework. Specifically, our goal is to compare the generalization gap of neural networks trained using the vanilla adversarial training method, which fully optimizes perturbations at every iteration, with the free adversarial training method, which simultaneously optimizes norm-bounded perturbations and classifier parameters. We prove bounds on the generalization error of these methods, indicating that the free adversarial training method may exhibit a lower generalization gap between training and test samples due to its simultaneous min-max optimization of classifier weights and perturbation variables. We conduct several numerical experiments to evaluate the train-to-test generalization gap in vanilla and free adversarial training methods. Our empirical findings also suggest that the free adversarial training method could lead to a smaller generalization gap over a similar number of training iterations. The paper code is available at `https://github.com/Xiwei-Cheng/Stability_FreeAT`.

## 1 Introduction

Deep neural networks (DNNs) have attained state-of-the-art results in various supervised learning tasks in computer vision, speech recognition, and natural language processing. However, it is widely recognized that DNNs are susceptible to minor adversarially designed perturbations to their input data, commonly regarded as *adversarial attacks* (Szegedy et al., 2013; Goodfellow et al., 2014). Adversarial examples are typically generated by optimizing a norm-constrained perturbation leading to the maximum classification loss for input data. A standard approach to defending against adversarial attacks is adversarial training (AT) (Madry et al., 2017), according to which the DNN classifier is learned using adversarially perturbed training data. AT methods have significantly improved the robustness of DNNs against norm-bounded perturbations. In recent years, several variants of AT methods have been developed to accelerate and facilitate the application of AT to large-scale models and datasets (Kannan et al., 2018; Shafahi et al., 2019; Wong et al., 2020).

While AT algorithms offer significant robustness against standard norm-bounded attacks, the generalization gap between their performance on training and test data has been found to be considerably greater than that of DNNs trained by standard empirical risk minimization (ERM) (Schmidt et al., 2018; Raghunathan et al., 2019). To study the overfitting phenomenon in adversarial training, the recent literature has focused

on the generalization analysis of adversarially trained models. The studies have attempted to analyze the generalization error in learning adversarially-robust models (Yin et al., 2019; Awasthi et al., 2020; Xiao et al., 2022b) and to show the reduction of generalization gap under explicit and implicit regularization methods such as early stopping and Lipschitz regularization (Rice et al., 2020; Zhang et al., 2022; Xiao et al., 2023b).

Specifically, several works (Lei et al., 2021; Farnia & Ozdaglar, 2021; Xiao et al., 2022b; 2024b) have concentrated on the interconnections between the optimization and generalization of adversarially-trained models. Since adversarial training methods use adversarial training examples with the worst-case norm-bounded perturbations, they are typically formulated as min-max optimization problems where the classifier and adversarial perturbations are the minimization and maximization variables, respectively. To solve the min-max optimization problem, the vanilla AT framework follows an iterative algorithm where, at every iteration, the inner maximization problem is fully solved for designing the optimal perturbations and subsequently, a single gradient update is applied to the DNN's parameters. Therefore, the vanilla AT results in a non-simultaneous optimization of the minimization and maximization variables of the underlying min-max problem. However, the theoretical generalization error bounds in (Farnia & Ozdaglar, 2021; Lei et al., 2021) suggest that the non-simultaneous optimization of the min and max variables in a min-max learning problem could lead to a greater generalization gap. Therefore, a natural question is whether an adversarial training algorithm with simultaneous optimization of the min and max problems can reduce the generalization gap.

In this work, we focus on a widely-used variant of adversarial training proposed by Shafahi et al. (2019), *adversarial training for free (Free AT)*, and aim to analyze its generalization behavior compared to the vanilla AT approach. While the vanilla AT follows a sequential optimization of the DNN and perturbation variables, the Free AT approach simultaneously computes the gradient of the two groups of variables at every round of applying the backpropagation algorithm to the multi-layer DNN. We aim to demonstrate that the simultaneous optimization of the classifier and adversarial examples in free AT could translate into a lower generalization error compared to vanilla AT. To this end, we provide theoretical and numerical results to compare the generalization properties of vanilla vs. free AT frameworks.

On the theory side, we leverage the algorithmic stability framework (Bousquet & Elisseeff, 2002; Hardt et al., 2015) to derive generalization error bounds for free and vanilla adversarial training methods. The shown generalization bounds suggest that in the nonconvex-nonconcave regime, the free AT algorithm could enjoy a lower generalization gap than the vanilla AT, since it applies simultaneous gradient updates to the DNN's and perturbations' variables. We also develop a similar generalization bound for the fast AT methodology (Goodfellow et al., 2014) which uses a single gradient step to optimize the perturbations. Our theoretical results suggest a comparable generalization bound between free and fast AT approaches.

Finally, we present the results of our numerical experiments to compare the generalization performance of the vanilla and free AT methods over standard computer vision datasets and neural network architectures. Our numerical results also suggest that the free AT method results in a considerably lower generalization gap than the vanilla AT. Furthermore, we propose applying the free AT framework to implement Free–TRADES, a free-AT version of the adversarial training algorithm TRADES (Zhang et al., 2019). The numerical results suggest that the TRADES algorithm similarly has a lower generalization gap under simultaneous optimization of the min and max optimization variables. This work's contributions can be summarized as:

- Applying the algorithmic stability framework to analyze the generalization behavior of free AT,

- Providing a theoretical comparison of the generalization properties of the vanilla and free AT methods,

- Numerically analyzing the generalization performance of the free vs. vanilla AT schemes under white-box and black-box adversarial attacks,

- Proposing Free–TRADES, a simultaneous updating variant of TRADES with better generalization.

## 2 Related Work

**Generalization in Adversarial Training:** Since the discovery of adversarial examples (Szegedy et al., 2013), a large body of works has focused on training robust DNNs against adversarial perturbations (Goodfellow et al., 2014; Carlini & Wagner, 2017; Madry et al., 2017; Zhang et al., 2019). Shafahi et al. (2019)

proposed the free adversarial training algorithm to update the neural net and adversarial perturbations simultaneously, aimed at reducing the computational cost of adversarial training. Also, Wong et al. (2020) discussed the application of the fast AT algorithm, aiming to lower the computational costs.

Compared to standard ERM training, the overfitting in adversarial training is shown to be more severe (Rice et al., 2020). A line of works analyzed adversarial generalization by applying uniform convergence tools such as VC-dimension (Montasser et al., 2019; Attias et al., 2022) and Rademacher complexity (Yin et al., 2019; Farnia et al., 2018; Awasthi et al., 2020; Xiao et al., 2022a; 2024a). Schmidt et al. (2018) proved tight bounds on the adversarially robust generalization error showing that vanilla adversarial training requires more data for proper generalization than standard training. Xing et al. (2022) studied the phase transition of generalization error from standard training to adversarial training. Also, the reference (Andriushchenko & Flammarion, 2020) discusses the catastrophic overfitting in the Fast AT method.

**Uniform Stability:** Bousquet & Elisseeff (2002) developed the algorithmic stability framework to analyze the generalization performance of learning algorithms. Hardt et al. (2015) further extended the algorithmic stability approach to stochastic gradient-based optimization (SGD) methods. Bassily et al. (2020); Lei (2023) analyzed the stability under non-smooth functions. Some recent works applied the stability framework to study the generalization gap of adversarial training, while they mostly assumed an oracle to obtain a perfect perturbation and focused on the stability of the training process. Xing et al. (2021) analyzed the stability by shedding light on the non-smooth nature of the adversarial loss. Xiao et al. (2022b) further investigated the stability bound by introducing a notion of approximate smoothness. Based on this result, Xiao et al. (2022c) proposed a smoothed version of SGDmax to improve the adversarial generalization. Xiao et al. (2023a) utilized the stability framework to improve the robustness of DNNs under various types of attacks.

**Generalization in minimax learning frameworks:** The generalization analysis of general minimax learning frameworks has been studied in several related works. Arora et al. (2017) established a uniform convergence generalization bound in terms of the discriminator's parameters in generative adversarial networks (GANs). Zhang et al. (2017); Bai et al. (2018) characterized the generalizability of GANs using the Rademacher complexity of the discriminator function space. Some work also analyzed generalization in GANs from the algorithmic perspective. Farnia & Ozdaglar (2021); Lei et al. (2021) compared the generalization of SGDA and SGDmax in minimax optimization problems using algorithmic stability. Wu et al. (2019) studied generalization in GANs from the perspective of differential privacy. Ozdaglar et al. (2022) proposed a new metric to evaluate the generalization of minimax problems and studied the generalization behaviors of SGDA and SGDmax.

## 3 Preliminaries

Suppose that labeled sample $(x, y)$ is randomly drawn from some unknown distribution $\mathcal{D}$. The goal of adversarial training is to find a model $f_w$ with parameter $w \in W$ which minimizes the population risk against the adversarial perturbation $\delta$ from a feasible perturbation set $\Delta$, defined as:

$$R(w) := \mathbb{E}_{(x,y) \sim \mathcal{D}} \left[ \max_{\delta \in \Delta} h(w, \delta; x, y) \right],$$

where $h(w, \delta; x, y) = \text{Loss}(f_w(x + \delta), y)$ is the loss function in the supervised learning problem. Since the learner does not have access to the underlying distribution $\mathcal{D}$ but only a dataset $S = \{x_1, x_2, \cdots, x_n\}$ of size $n$, we define the empirical adversarial risk as

$$R_S(w) := \frac{1}{n} \sum_{j=1}^{n} \max_{\delta \in \Delta} h(w, \delta; x_j, y_j).$$

The generalization adversarial risk $\mathcal{E}_{\text{gen}}(w)$ of model parameter $w$ is defined as the difference between population and empirical risk, i.e., $\mathcal{E}_{\text{gen}}(w) := R(w) - R_S(w)$. For a potentially randomized algorithm $A$ which takes a dataset $S$ as input and outputs a random vector $w = A(S)$, we can define its expected generalization adversarial risk over the randomness of a training set $S$ and stochastic algorithm $A$, e.g. under mini-batch

selection in stochastic gradient methods or random initialization of the weights of a neural net classifier,

$$\mathcal{E}_{\text{gen}}(A) := \mathbb{E}_{S,A}\big[R(A(S)) - R_S(A(S))\big].$$

Throughout the paper, unless specified otherwise, we use $\|\cdot\|$ to denote the $\mathcal{L}_2$ norm of vectors or the Frobenius norm of matrices.

### 3.1 Adversarial Training

In standard applications of adversarial training, the perturbation set $\Delta$ is usually an $\mathcal{L}_2$-norm or $\mathcal{L}_\infty$-norm bounded ball of some small radius $\varepsilon$ (Szegedy et al., 2013; Goodfellow et al., 2014). To robustify a neural network, the standard methodology $A_{\text{Vanilla}}$ is to train the network with (approximately) perfectly perturbed samples, both in practice (Madry et al., 2017; Rice et al., 2020) and in theoretical analysis (Xing et al., 2021; Xiao et al., 2022b), which is formally defined as follows:

---

**Algorithm 1** Vanilla Adversarial Training Algorithm $A_{\text{Vanilla}}$

---
1: **Input:** Training samples $S$, perturbation set $\Delta$, learning rate of model weight $\alpha_w$, mini-batch size $b$, number of iterations $T$
2: **for** step $t \leftarrow 1, \cdots, T$ **do**
3:     Uniformly random mini-batch $B \subset S$ of size $b$
4:     Compute adversarial attack $\delta_j$ for all $(x_j, y_j) \in B$: $\delta_j \leftarrow \arg\max_{\tilde{\delta} \in \Delta} h(w, \tilde{\delta}; x_j, y_j)$
5:     Update $w$ with perturbed samples: $w \leftarrow w - \frac{\alpha_w}{b} \sum_{(x_j, y_j) \in B} \nabla_w h(w, \delta_j; x_j, y_j)$
6: **end for**

---

In practice, due to the non-convexity of neural networks, it is computationally intractable to compute the best adversarial attack $\delta = \arg\max_{\tilde{\delta} \in \Delta} h(w, \tilde{\delta}; x)$, but the standard projected gradient descent (PGD) attack (Madry et al., 2017) is widely believed to produce near-optimal attacks, by iteratively projecting the gradient $\nabla_\delta h(w, \delta; x)$ onto the set of extreme points of $\Delta$, i.e.,

$$\pi_\Delta(g) := \underset{\tilde{\delta} \in \text{ExtremePoints}(\Delta)}{\arg\min} \|g - \tilde{\delta}\|^2, \tag{1}$$

updating the attack $\delta$ with the projected gradient $\pi_\Delta(\nabla_\delta h(w, \delta; x))$ and some step size $\alpha_\delta$, and projecting the update attack to the feasible set $\Delta$, i.e,

$$\mathcal{P}_\Delta(g) := \underset{\delta \in \Delta}{\arg\min} \|g - \delta\|^2. \tag{2}$$

Despite the significant robustness gained from $A_{\text{Vanilla}}$, it demands significant computational costs for training. The Free adversarial training algorithm $A_{\text{Free}}$ (Shafahi et al., 2019) is proposed to avoid the overhead cost, by simultaneously updating the model weight parameter $w$ when performing PGD attacks. $A_{\text{Free}}$ is empirically observed to achieve comparable robustness to $A_{\text{Vanilla}}$, while it can considerably reduce the training time (Shafahi et al., 2019; Wong et al., 2020).

---

**Algorithm 2** Free Adversarial Training Algorithm $A_{\text{Free}}$

---
1: **Input:** Training samples $S$, perturbation set $\Delta$, step size of model weight $\alpha_w$, learning rate of adversarial attack $\alpha_\delta$, free step $m$, mini-batch size $b$, number of iterations $T$
2: **for** step $\leftarrow 1, \cdots, T/m$ **do**
3:     Uniformly random mini-batch $B \subset S$ of size $b$
4:     $\delta := [\delta_j]_{\{j:(x_j,y_j)\in B\}} \leftarrow \text{Uniform}(\Delta^b)$
5:     **for** iteration $i \leftarrow 1, \cdots, m$ **do**
6:         Compute weight gradient and attack gradient by backpropagation:
7:         $g_w \leftarrow \frac{1}{b} \sum_{(x_j,y_j)\in B} \nabla_w h(w, \delta_j; x_j, y_j)$, and $g_\delta \leftarrow [\nabla_\delta h(w, \delta_j; x_j, y_j)]_{\{j:(x_j,y_j)\in B\}}$
8:         Update $w$ with mini-batch gradient descent: $w \leftarrow w - \alpha_w g_w$
9:         Update $\delta$ with projected gradient ascent: $\delta \leftarrow [\mathcal{P}_\Delta(\delta_j + \alpha_\delta \pi_\Delta(g_{\delta_j}))]_{\{j:(x_j,y_j)\in B\}}$
10:     **end for**
11: **end for**

---

We also compare $A_{\text{Vanilla}}$ and $A_{\text{Free}}$ with the Fast adversarial training algorithm $A_{\text{Fast}}$ (Wong et al., 2020), a variant of the fast gradient sign method (FGSM) proposed by Goodfellow et al. (2014). Instead of computing a perfect perturbation, it applies only one projected gradient step with fine-tuned step size from a randomly initialized point in $\Delta$. $A_{\text{Fast}}$ is empirically shown to achieve comparable robustness to the standard PGD training with lower training costs (Wong et al., 2020; Andriushchenko & Flammarion, 2020).

---

**Algorithm 3** Fast Adversarial Training Algorithm $A_{\text{Fast}}$

---
1: **Input:** Training samples $X$, perturbation set $\Delta$, , learning rate of model weight $\alpha_w$, step size of adversarial attack $\tilde{\alpha}_\delta$, mini-batch size $b$, number of iterations $T$
2: **for** step $t \leftarrow 1, \cdots, T$ **do**
3:     Uniformly random mini-batch $B \subset S$ of size $b$
4:     Compute adversarial attack $\delta$ with random start:
5:     $\tilde{\delta} := [\tilde{\delta}_j]_{\{j:(x_j,y_j)\in B\}} \leftarrow \text{Uniform}(\Delta^b)$
6:     $g_\delta \leftarrow [\nabla_\delta h(w, \tilde{\delta}_j; x_j, y_j)]_{\{j:(x_j,y_j)\in B\}}$
7:     $\delta \leftarrow [\mathcal{P}_\Delta(\tilde{\delta}_j + \alpha_\delta \pi_\Delta(g_{\delta_j}))]_{\{j:(x_j,y_j)\in B\}}$
8:     Update $w$ with perturbed sample: $w \leftarrow w - \frac{\alpha_w}{b} \sum_{(x_j,y_j)\in B} \nabla_w h(w, \delta_j; x_j, y_j)$
9: **end for**

---

## 4 Stability and Generalization in Adversarial Training

To bound the generalization adversarial risk, the notion of uniform stability with respect to the adversarial loss is introduced (Bousquet & Elisseeff, 2002).

**Definition 1.** *A randomized algorithm $A$ is $\epsilon$-uniformly stable if for all datasets $S, S' \in \mathcal{D}^n$ such that $S$ and $S'$ differ in at most one example, we have*

$$\sup_x \mathbb{E}_A \left[ \max_{\delta \in \Delta} h(A(S), \delta; x) - \max_{\delta \in \Delta} h(A(S'), \delta; x) \right] \le \epsilon. \tag{3}$$

Similar to Theorem 2.2 in Hardt et al. (2015), the generalization risk in expectation of a uniformly stable algorithm can be bounded by the following theorem

**Theorem 1.** *Assume that a randomized algorithm $A$ is $\epsilon$-uniformly stable, then the expected generalization risk satisfies*

$$|\mathcal{E}_{gen}| = |\mathbb{E}_{S,A}[R(A(S)) - R_S(A(S))]| \le \epsilon.$$

*Proof.* The proof can be found in Theorem 2.2 in Hardt et al. (2015) by replacing the loss function with the adversarial loss $\max_{\delta \in \Delta} h(w, \delta; x)$. $\qquad \square$

In order to study the uniform stability of adversarial training, we make the following assumptions on the Lipschitzness and smoothness of the objective function. Our generalization results will hold as long as Assumptions 1, 2 hold locally within an attack radius distance from the support set of $X$.

**Assumption 1.** *$h(w, \delta)$ is jointly $L$-Lipschitz in $(w, \delta)$ and $L_w$-Lipschitz in $w$ over $W \times \Delta$, i.e., for every $w, w' \in W$ and $\delta, \delta' \in \Delta$ we have*

$$|h(w, \delta) - h(w', \delta')|^2 \le L^2 \left( \|w - w'\|^2 + \|\delta - \delta'\|^2 \right),$$
$$|h(w, \delta) - h(w', \delta)|^2 \le L_w^2 \|w - w'\|^2.$$

**Assumption 2.** *$h(w, \delta)$ is continuously differentiable and $\beta$-smooth over $W \times \Delta$, i.e., $[\nabla_w h(w, \delta), \nabla_\delta h(w, \delta)]$ is $\beta$-Lipschitz over $W \times \Delta$ and for every $w, w' \in W$, $\delta, \delta' \in \Delta$ we have*

$$\|\nabla_w h(w, \delta) - \nabla_w h(w', \delta')\|^2 + \|\nabla_\delta h(w, \delta) - \nabla_\delta h(w', \delta')\|^2$$
$$\le \beta^2 \left( \|w - w'\|^2 + \|\delta - \delta'\|^2 \right).$$

We clarify that the Lipschitzness and smoothness assumptions are common practice in the uniform stability analysis (Hardt et al., 2015; Xing et al., 2021; Farnia & Ozdaglar, 2021; Xiao et al., 2022b). In practice, although ReLU activation function is non-smooth, recent works (Du et al., 2019; Allen-Zhu et al., 2019) showed that the loss function of over-parameterized neural networks is semi-smooth; also, another line of works (Xie et al., 2020; Singla et al., 2021) suggest that replacing ReLU with smooth activation functions can strengthen adversarial training; and some works (Fazlyab et al., 2019; Shi et al., 2022) attempt to compute the Lipschitz constant of neural networks.

## 5 Stability-based Generalization Bounds for Free AT

In this section, we provide generalization bounds on vanilla, free, and fast adversarial training algorithms. While previous works mainly focus on theoretically analyzing the stability behaviors of vanilla adversarial training under the scenario that $h(w, \delta; x)$ is convex in $w$ (Xing et al., 2021; Xiao et al., 2022b), or $h(w, \delta; x)$ is concave or even strongly-concave in $\delta$ (Lei et al., 2021; Farnia & Ozdaglar, 2021; Yang et al., 2022; Ozdaglar et al., 2022), our analysis focuses on the nonconvex-nonconcave scenario: without assumptions on the convexity of $h(w, \delta; x)$ in $w$ or concavity of $h(w, \delta; x)$ in $\delta$. We defer the proof of Theorems 2 and 4 to the Appendix A.1 and A.2. Throughout the proof, we assume that Assumptions 1 and 2 hold.

**Theorem 2** (Stability generalization bound of $A_{\text{Vanilla}}$). *Assume that $h(w, \delta)$ satisfies Assumptions 1 and 2 and is bounded in $[0, 1]$, and the perturbation set is an $\mathcal{L}_2$-norm ball of some constant radius $\varepsilon$, i.e., $\Delta = \{\delta : ||\delta|| \leq \varepsilon\}$. Suppose that we run $A_{Vanilla}$ in Algorithm 1 for $T$ steps with vanishing step size $\alpha_{w,t} \leq c/t$. Let constant $\lambda_{Vanilla} := \beta c$, then*

$$\mathcal{E}_{gen}(A_{Vanilla}) \leq \frac{b}{n}\left(1 + \frac{1}{\lambda_{Vanilla}}\right)\left(\frac{2L_w c}{b}(\varepsilon \beta n + L)\right)^{\frac{1}{\lambda_{Vanilla}+1}} T^{\frac{\lambda_{Vanilla}}{\lambda_{Vanilla}+1}}. \tag{4}$$

By equation 4, we have the following asymptotic bound on $\mathcal{E}_{\text{gen}}(A_{\text{Vanilla}})$ with respect to $T$ and $n$

$$\mathcal{E}_{\text{gen}}(A_{\text{Vanilla}}) = \mathcal{O}\left(T^{\frac{\lambda_{\text{Vanilla}}}{\lambda_{\text{Vanilla}}+1}}/n^{\frac{\lambda_{\text{Vanilla}}}{\lambda_{\text{Vanilla}}+1}}\right). \tag{5}$$

This bound suggests that the vanilla adversarial training algorithm could lead to large generalization gaps, because for any $T = \Omega(n)$, the bound $T^{\frac{\lambda_{\text{Vanilla}}}{\lambda_{\text{Vanilla}}+1}}/n^{\frac{\lambda_{\text{Vanilla}}}{\lambda_{\text{Vanilla}}+1}} = \Omega(1)$ is non-vanishing even when we are given infinity samples. This implication is also confirmed by the following lower bound from the work of Xing et al. (2021) and Xiao et al. (2022b):

**Theorem 3** (Lower bound on stability; Theorem 1 in Xing et al. (2021), Theorem 5.2 in Xiao et al. (2022b)). *Suppose $\Delta = \{\delta : ||\delta|| \leq \varepsilon\}$. Assume $w(S)$ is the output of running $A_{Vanilla}$ on the dataset $S$ with mini-batch size $b = 1$ and constant step size $\alpha_w \leq 1/\beta$ for $T$ steps. There exist some loss function $h(w, \delta; x)$ which is differentiable and convex with respect to $w$, some constant $\varepsilon > 0$, and some datasets $S$ and $S'$ that differ in only one sample, such that*

$$\mathbb{E}[||w(S) - w(S')||] \geq \Omega\left(\sqrt{T} + \frac{T}{n}\right). \tag{6}$$

This lower bound indicates that $A_{\text{Vanilla}}$ could lack stability when the attack radius $\varepsilon = \Omega(1)$, hence the algorithm may result in significant generalization error from the stability perspective. Note that the lower bound in equation 6 is not inconsistent with Theorem 2, in which the step-size is assumed to be vanishing $\alpha_{w,t} \leq c/t$ and thus the lower bound is not directly applicable under that assumption. However, this constant generalization gap could be reduced by free adversarial training.

**Theorem 4** (Stability generalization bound of $A_{\text{Free}}$). *Assume that $h(w, \delta)$ satisfies Assumptions 1 and 2 and is bounded in $[0, 1]$, and the perturbation set is an $\mathcal{L}_2$-norm ball of some constant radius $\varepsilon$, i.e., $\Delta = \{\delta : ||\delta|| \leq \varepsilon\}$. Suppose we run $A_{Free}$ in Algorithm 2 for $T/m$ steps with vanishing step size $\alpha_{w,t} \leq c/mt$ and constant step size $\alpha_\delta$. If the norm of gradient $\nabla_\delta h(w, \delta; x)$ is lower bounded by $1/\psi$ for constant $\psi > 0$ with probability 1 during the training process, let constant $\lambda_{Free} := \beta c(1 + \beta c/m + \alpha_\delta \varepsilon \psi \beta)^{m-1}$, then*

$$\mathcal{E}_{gen}(A_{Free}) \leq \frac{b}{n}\left(1 + \frac{1}{\lambda_{Free}}\right)\left(\frac{2LL_w}{b\beta}\lambda_{Free}\right)^{\frac{1}{\lambda_{Free}+1}}\left(\frac{T}{m}\right)^{\frac{\lambda_{Free}}{\lambda_{Free}+1}}. \tag{7}$$

**Remark 1.** *To validate the soundness of the assumption on the lower-bounded norm $\nabla_\delta h(w, \delta; x)$ by constant $1/\psi$ in practical settings, in Appendix B.6 we present the numerical evaluation of the gradient norm over the course of free-AT on CIFAR-10 and CIFAR-100 data, indicating that the norm is consistently lower-bounded by $\mathcal{O}(10^{-3})$ in the experiments.*

**Remark 2.** *Theorem 4 indicates how the simultaneous updates influence the generalization of adversarial training. From equation 7, we have the following asymptotic bound on $\mathcal{E}_{gen}(A_{Free})$ with respect to $T$ and $n$*

$$\mathcal{E}_{gen}(A_{Free}) = \mathcal{O}\left(T^{\frac{\lambda_{Free}}{\lambda_{Free}+1}}/n\right). \tag{8}$$

*Therefore, by controlling the step size $\alpha_\delta$ of the maximization step, we can bound the coefficient $\lambda_{Free}$ and thus control the generalization gap of $A_{Free}$, where a lower $\alpha_\delta$ can result in a smaller generalization gap.*

**Remark 3.** *One technical contribution of this work is to perform the stability analysis where the maximization variable is re-initialized after every m iterations where a new mini-batch of data is used. Our theoretical results suggest that as long as m is bounded by $\mathcal{O}(\alpha_\delta \epsilon \psi/c)$, the generalization risk will not change significantly with a greater m value, which is not implied by the standard bounds in the existing works (Farnia & Ozdaglar, 2021) in the literature. Also, our theoretical analysis considers the normalized gradient (instead of the vanilla gradient) for the gradient ascent step of solving the maximization sub-problem and mini-batch stochastic optimization for updating min and max variables at every iteration, which are not analyzed in the previous literature (Farnia & Ozdaglar, 2021).*

We note that Theorem 3 (Xing et al., 2021; Xiao et al., 2022b) gives a lower bound $\Omega(T/n)$ on the algorithmic stability of vanilla adversarial training. On the other hand, comparing equation 8 with equation 5 suggests that $A_{\text{Free}}$ can generalize better than $A_{\text{Vanilla}}$ for any $T = \mathcal{O}(n)$, since

$$\frac{T^{\frac{\lambda_{\text{Free}}}{\lambda_{\text{Free}}+1}}/n}{T^{\frac{\lambda_{\text{Vanilla}}}{\lambda_{\text{Vanilla}}+1}}/n^{\frac{\lambda_{\text{Vanilla}}}{\lambda_{\text{Vanilla}}+1}}} = \left(\frac{T}{n}\right)^{\frac{1}{\lambda_{\text{Vanilla}}+1}} \left(\frac{1}{T}\right)^{\frac{1}{\lambda_{\text{Free}}+1}} = \mathcal{O}\left(1/T^{\frac{1}{\lambda_{\text{Free}}+1}}\right).$$

Furthermore, when $T = \mathcal{O}(n)$, equation 8 gives $\mathcal{E}_{\text{gen}}(A_{\text{Free}}) = \mathcal{O}\left(1/n^{\frac{1}{\lambda_{\text{Free}}+1}}\right)$, which implies that the generalization gap of $A_{\text{Free}}$ can be bounded given enough samples. If the number of iterations $T$ is fixed, one can see that the generalization gap of $A_{\text{Free}}$ has a faster convergence to 0 than $A_{\text{Vanilla}}$. Therefore, neural nets trained by the free AT algorithm could generalize better than the vanilla adversarially-trained networks due to their improved algorithmic stability. Our theoretical results also echo the conclusion in Schmidt et al. (2018) that adversarially robust generalization requires more data, since $\lambda_{\text{Free}}$ increases with respect to $\varepsilon$.

**Remark 4.** *We note that the Free-AT method in Algorithm 2 follows the update rule of a projected gradient descent ascent (Projected GDA) which has been widely studied in the optimization literature. To the best of our knowledge, a tight convergence rate for Projected GDA applied to a general nonconvex-nonconcave optimization problem is still an open question in the community (Lin et al., 2020; Li et al., 2022). As a tight convergence rate for GDA nonconvex-nonconcave optimization is not available in the literature, we relied on numerical experiments to verify the improvement in the generalization gap by Free-AT method. Note that our numerical experiments use standard selections of stepsizes and other hyperparameters for the AT algorithms, and their numerical results suggest the standard hyperparameter selection leads to a lower generalization gap for Free-AT compared to Vanilla-AT.*

We also provide theoretical analysis for the fast adversarial training algorithm $A_{\text{Fast}}$ in Theorem 5, whose proof is deferred to Appendix A.3.

**Theorem 5** (Stability generalization bound of $A_{\text{Fast}}$). *Assume that $h(w, \delta)$ satisfies Assumptions 1 and 2 and is bounded in $[0, 1]$, and the perturbation set is an $\mathcal{L}_2$-norm ball of some constant radius $\varepsilon$, i.e., $\Delta = \{\delta : ||\delta|| \leq \varepsilon\}$. Suppose that we run $A_{\text{Fast}}$ in Algorithm 3 for $T$ steps with vanishing step size $\alpha_{w,t} \leq c/t$ and constant step size $\tilde{\alpha}_\delta$. If the norm of gradient $\nabla_\delta h(w, \delta; x)$ is lower bounded by $1/\psi$ for some constant $\psi > 0$ with probability 1 during the training process, let constant $\lambda_{Fast} := \beta c(1 + \tilde{\alpha}_\delta \varepsilon \psi \beta)$, then*

$$\mathcal{E}_{gen}(A_{Fast}) \leq \frac{b}{n}\left(1 + \frac{1}{\lambda_{Fast}}\right)\left(\frac{2cLL_w}{b}\right)^{\frac{1}{\lambda_{Fast}+1}} T^{\frac{\lambda_{Fast}}{\lambda_{Fast}+1}}. \tag{9}$$

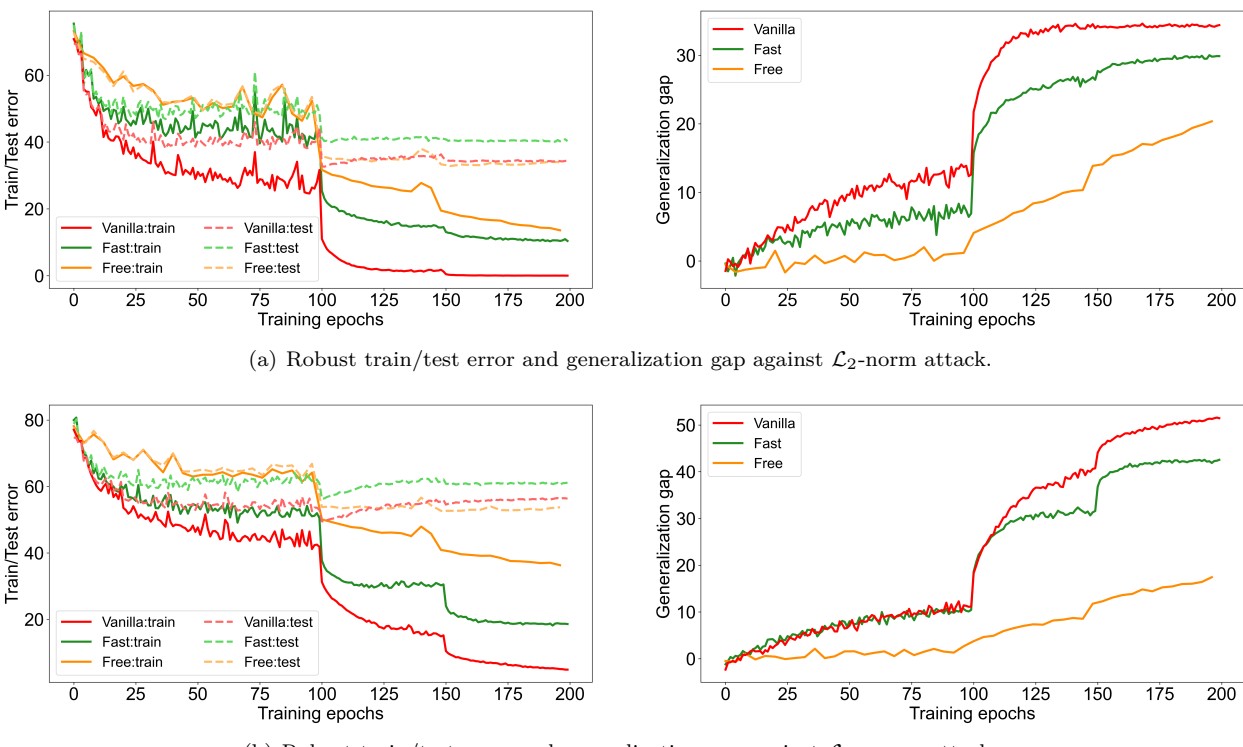

(a) Robust train/test error and generalization gap against $\mathcal{L}_2$-norm attack.

(b) Robust train/test error and generalization gap against $\mathcal{L}_\infty$-norm attack.

Figure 1: Learning curves of different algorithms for a ResNet18 model adversarially trained against $\mathcal{L}_2$ and $\mathcal{L}_\infty$ attacks on CIFAR-10. The free curves are scaled horizontally by a factor of $m$.

Similar to Remark 2, we have the following asymptotic bound on $\mathcal{E}_{\text{gen}}(A_{\text{Fast}})$ with respect to $T$ and $n$

$$\mathcal{E}_{\text{gen}}(A_{\text{Fast}}) = \mathcal{O}\left(T^{\frac{\lambda_{\text{Fast}}}{\lambda_{\text{Fast}}+1}}/n\right). \tag{10}$$

Therefore, by controlling the step size $\tilde{\alpha}_\delta$ of the maximization step, we can bound the coefficient $\lambda_{\text{Fast}}$ and thus control the generalization gap of $A_{\text{Fast}}$, where a lower $\tilde{\alpha}_\delta$ can result in a smaller generalization gap. Comparing equation 10 with equation 5 also suggests that for any $T = \mathcal{O}(n)$, $A_{\text{Fast}}$ can generalize better than $A_{\text{Vanilla}}$, since $\frac{T^{\frac{\lambda_{\text{Fast}}}{\lambda_{\text{Fast}}+1}}/n}{T^{\frac{\lambda_{\text{Vanilla}}}{\lambda_{\text{Vanilla}}+1}}/n^{\frac{\lambda_{\text{Vanilla}}}{\lambda_{\text{Vanilla}}+1}}} = \left(\frac{T}{n}\right)^{\frac{1}{\lambda_{\text{Vanilla}}+1}}\left(\frac{1}{T}\right)^{\frac{1}{\lambda_{\text{Fast}}+1}} = \mathcal{O}\left(1/T^{\frac{1}{\lambda_{\text{Fast}}+1}}\right)$.

# 6 Numerical Results

In this section, we evaluate the generalization performance of vanilla, fast, and free adversarial training algorithms in a series of numerical experiments. We first demonstrate the overfitting issue in vanilla adversarial training and show that free or fast algorithms can considerably reduce the generalization gap. We demonstrate that the smaller generalization gap could translate into greater robustness against score-based or transferred black-box attacks. To examine the advantages of free AT, we also study the generalization gap for different numbers of training samples.

**Experiment Settings:** We conduct our experiments on datasets CIFAR-10, CIFAR-100 (Krizhevsky & Hinton, 2009), Tiny-ImageNet (Le & Yang, 2015), and SVHN (Netzer et al., 2011). Following the standard setting in Madry et al. (2017), we use ResNet18 (He et al., 2016) for CIFAR-10 and CIFAR-100, ResNet50 for Tiny-ImageNet, and VGG19 (Simonyan & Zisserman, 2014) for SVHN to validate our results on a diverse selection of network architectures. For vanilla adversarial training algorithm, since the inner optimization

Table 1: Robust training accuracy, testing accuracy, and generalization gap of the vanilla, fast, and free algorithms across CIFAR-10 and CIFAR-100 datasets.

| Dataset | Attack | Results (%) | Vanilla-7 | Vanilla-10 | Fast | Free-2 | Free-4 | Free-6 | Free-8 | Free-10 |
|---|---|---|---|---|---|---|---|---|---|---|
| CIFAR-10 | $\mathcal{L}_2$ | Train Acc. | 100.0 | 100.0 | 89.7 | 89.5 | 86.4 | 78.9 | 74.3 | 71.1 |
| | | Test Acc. | 65.5 | 65.6 | 59.7 | 62.1 | 66.0 | 66.2 | 65.4 | 64.2 |
| | | Gen. Gap | 34.5 | 34.4 | 30.0 | 27.4 | 20.4 | 12.7 | 8.9 | 6.9 |
| | $\mathcal{L}_\infty$ | Train Acc. | 94.8 | 95.1 | 81.4 | 37.9 | 63.7 | 58.8 | 55.6 | 52.7 |
| | | Test Acc. | 42.9 | 43.7 | 38.9 | 28.2 | 46.3 | 47.2 | 48.2 | 46.9 |
| | | Gen. Gap | 51.9 | 51.4 | 42.5 | 9.7 | 17.4 | 11.6 | 7.4 | 5.8 |
| CIFAR-100 | $\mathcal{L}_2$ | Train Acc. | 99.9 | 99.9 | 81.5 | 94.0 | 82.9 | 68.1 | 58.2 | 49.7 |
| | | Test Acc. | 35.7 | 36.5 | 30.8 | 34.8 | 36.6 | 38.0 | 37.8 | 36.9 |
| | | Gen. Gap | 64.2 | 63.4 | 50.7 | 59.2 | 46.3 | 30.1 | 20.4 | 12.8 |
| | $\mathcal{L}_\infty$ | Train Acc. | 95.6 | 96.0 | 70.3 | 36.9 | 52.9 | 43.2 | 36.7 | 32.2 |
| | | Test Acc. | 20.0 | 20.3 | 17.3 | 15.0 | 22.4 | 24.5 | 24.9 | 24.5 |
| | | Gen. Gap | 75.6 | 75.7 | 53.0 | 21.9 | 30.5 | 18.7 | 11.8 | 7.7 |

task $\max_{\delta \in \Delta} h(w, \delta; x)$ is computationally intractable for neural networks which are generally non-concave, we apply standard projected gradient descent (PGD) attacks (Madry et al., 2017) as a surrogate adversary. For free and fast algorithms, we adopt $A_{\text{Free}}$ and $A_{\text{Fast}}$ defined in Algorithms 2 and 3, following from Shafahi et al. (2019); Wong et al. (2020).

**Robust Overfitting during Training Process:** We applied $\mathcal{L}_2$-norm attack of radius $\varepsilon = 128/255$ and $\mathcal{L}_\infty$-norm attack of radius $\varepsilon = 8/255$ to adversarially train ResNet18 models on CIFAR-10. For the vanilla algorithm, we used a PGD adversary to perturb the image. For the free algorithm, we applied the learning rate of adversarial attack $\alpha_\delta = \varepsilon$ with free step $m$ as 2, 4, 6, 8, and 10.[1] The other implementation details are deferred to Appendix B.1. We trained the models for 200 epochs and after every epoch, we tested the models' robust accuracy against a PGD adversary and evaluated the generalization gap. The numerical results on CIFAR-10 and CIFAR-100 are presented in Table 1, and the training curves are plotted in Figure 1. Further numerical results on SVHN and Tiny-ImageNet are deferred to Table 4 in Appendix B.1.

Based on the empirical results, we observe the significant overfitting in the robust accuracy of the vanilla adversarial training: the generalization gap is above 30% against $\mathcal{L}_2$ attack and 50% against $\mathcal{L}_\infty$ attack. On the other hand, the free AT algorithm has less severe overfitting and reduced the generalization gap to 20%. Although the free AT algorithm applies a weaker adversary, it achieves comparable robustness on test samples to the vanilla AT algorithm against the PGD attacks by lowering the generalization gap. Additional numerical results for different numbers of free AT steps and on other datasets are provided in Appendix B.1.

**Robustness Evaluation Against Black-box Attacks:** To study the consequences of the generalization behavior of the free AT algorithm, we evaluated the robustness of the adversarially-trained networks against black-box attack schemes where the attacker does not have access to the parameters of the target models (Bhagoji et al., 2018). We applied the square attack (Andriushchenko et al., 2020), a score-based methodology via random search, to examine networks adversarially trained by the discussed algorithms as shown in Figure 2. We also used adversarial examples transferred from other independently trained robust models as shown in Figure 3. More experiments on different datasets are provided in Appendix B.2.

We extensively observe the improvements of the free algorithm compared to the vanilla algorithm against different black-box attacks, which suggests that its robustness is not gained from gradient-masking (Athalye et al., 2018) but rather attributed to the smaller generalization gap. Furthermore, our numerical findings in Appendix B.3 indicate that the adversarial perturbations designed for DNNs trained by free AT could transfer better to an unseen target neural net than those found for DNNs trained by vanilla and fast AT.

**Generalization Gap for Different Numbers of Training Samples:** To examine our theoretical results in Theorems 2 and 4, we evaluated the robust generalization loss with respect to different num-

---

[1]Throughout this work, we use "Vanilla-$m$" to denote the vanilla AT algorithm with $m$ PGD-adversary iterations, and "Vanilla" without specification means Vanilla-10 by default. We also use "Free-$m$" to denote the free AT algorithm with $m$ free steps, and "Free" without specification means Free-4 by default. It is worth noting that Vanilla-1 is equivalent to fast AT algorithm.

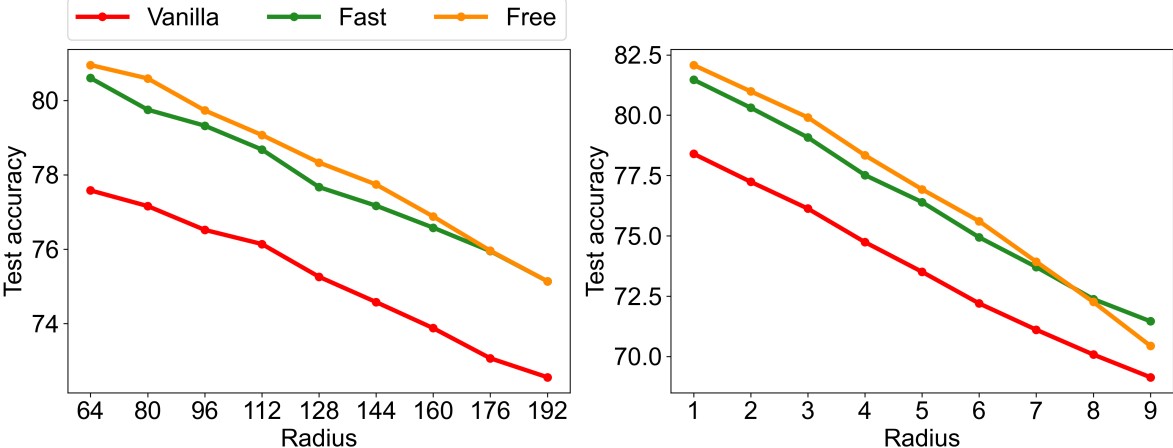

Figure 2: Robust accuracy of ResNet18 models adversarially trained by vanilla, fast, and free algorithms against square attack on CIFAR10. The left figure applies $\mathcal{L}_2$ attacks of radius ranging from 64 to 192, and the right figure applies $\mathcal{L}_\infty$ attacks of radius ranging from 1 to 9.

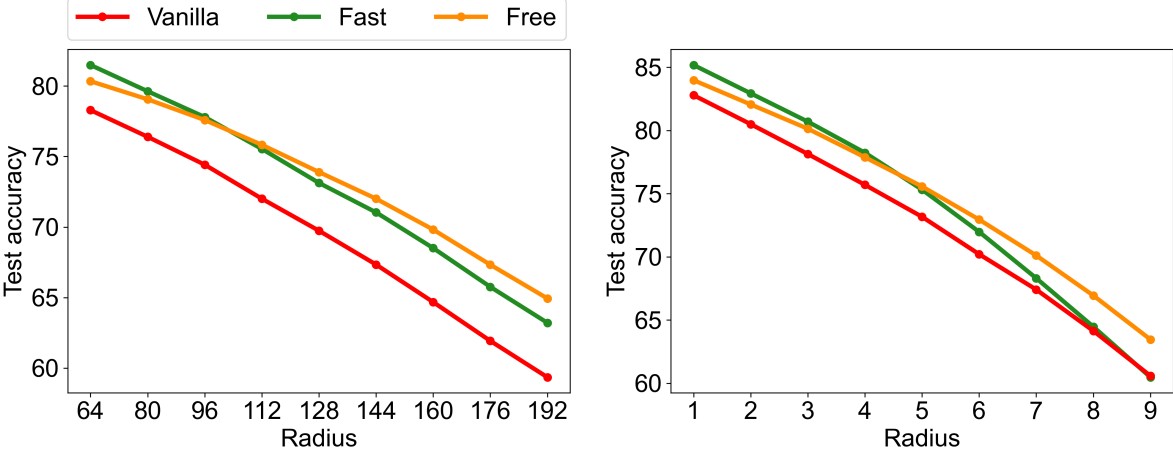

Figure 3: Robust accuracy against transferred attacks designed for another independently trained robust model. The left figure applies $\mathcal{L}_2$ attacks of radius ranging from 64 to 192, and the right figure applies $\mathcal{L}_\infty$ attacks of radius ranging from 1 to 9.

bers of training samples $n$. We randomly sampled a subset from the CIFAR-10 training dataset of size $n \in \{10000, 20000, 30000, 40000, 50000\}$, and adversarially trained ResNet18 models on the subset for a fixed number of iterations. As shown in Figure 4, the generalization gap of free AT is notably decreasing faster than vanilla AT with respect to $n$, which is consistent with our theoretical analysis. More experimental results are discussed in the Appendix B.4.

**Generalization analysis of friendly adversarial training (Zhang et al., 2020) with adaptive attack steps:** We also perform numerical experiments to compare vanilla AT with friendly adversarial training (Friendly-AT) (Zhang et al., 2020), which applies an adaptive number of steps and an adaptive training perturbation radius $\varepsilon_{\text{train}}$. As shown in Table 2, by choosing a sufficiently large $\varepsilon_{\text{train}}$ its generalization improvement over vanilla AT is similar to that of Fast-AT. On the other hand, a smaller $\varepsilon_{\text{train}}$ results in PGD AT-like numerical results, showing the trade-off in generalization-optimization accuracy explored by Friendly-AT. The theoretical analysis of Friendly-AT will be an interesting future direction for our work.

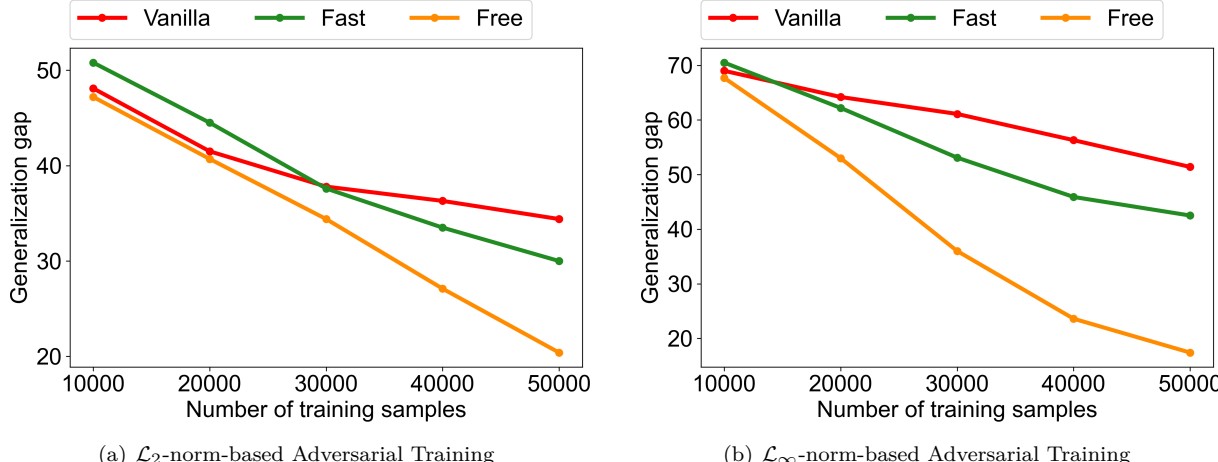

(a) $\mathcal{L}_2$-norm-based Adversarial Training

(b) $\mathcal{L}_\infty$-norm-based Adversarial Training

Figure 4: Adversarial generalization gap of ResNet18 models adversarially trained by vanilla, fast, and free algorithm for a fixed number of steps on a subset of CIFAR-10.

Table 2: Robust generalization performance of Friendly-AT (Zhang et al., 2020) and Vanilla-AT for ResNet18 models adversarially trained against $\mathcal{L}_2$ and $\mathcal{L}_\infty$ attacks on CIFAR-10. We set the extent steps $\tau = 3$ in Friendly-AT, and other implementations follow from Zhang et al. (2020).

| Results (%) | CIFAR-10, $\mathcal{L}_2$ attack | | | CIFAR-10, $\mathcal{L}_\infty$ attack | | |
| | Vanilla | Friendly $\varepsilon_{\text{train}} = 128/255$ | Friendly $\varepsilon_{\text{train}} = 256/255$ | Vanilla | Friendly $\varepsilon_{\text{train}} = 8/255$ | Friendly $\varepsilon_{\text{train}} = 16/255$ |
| --- | --- | --- | --- | --- | --- | --- |
| Train Acc. | 100.0 | 99.9 | 100.0 | 95.1 | 94.7 | 88.7 |
| Test Acc. | 65.6 | 65.8 | 66.9 | 43.7 | 44.1 | 46.3 |
| Gen. Gap | 34.4 | 34.1 | 33.1 | 51.4 | 50.6 | 42.4 |

# 7 Free–TRADES

Our theoretical results suggest that the improved generalization in the free AT algorithm could follow from its simultaneous min-max optimization updates. A natural question is whether we can extend these results to other adversarial training methods. Here we propose the Free–TRADES algorithm, a combination of the free AT algorithm and another well-established adversarial learning algorithm, TRADES (Zhang et al., 2019), and numerically evaluate the proposed Free–TRADES method.

The main characteristic of TRADES can be summarized as substituting the adversarial loss $h(w, \delta; x, y) = \text{Loss}(f_w(x + \delta), y)$ with a surrogate loss:

$$\tilde{h}_\lambda(w, \delta; x, y) := \text{Loss}(f_w(x), y) + \frac{1}{\lambda}\text{Loss}(f_w(x), f_w(x + \delta)). \tag{11}$$

The TRADES algorithm is aimed to minimize the surrogate risk $\frac{1}{n}\sum_{j=1}^{n}\max_{\delta \in \Delta}\tilde{h}_\lambda(w, \delta; x_j, y_j)$. Therefore, a natural idea gained from our theoretical analysis is to apply simultaneous updates to the adversarial attack $\delta$ and model weight $w$, which is stated in Algorithm 4.

We performed several numerical experiments to compare the performance of TRADES and Free–TRADES algorithms. The results demonstrated in Table 3 show that Free–TRADES could considerably improve the generalization gap while attaining a comparable (sometimes better) test performance to TRADES, which indicates that other adversarial training algorithms different from vanilla AT can also benefit from simultaneous optimization updates. Furthermore, we note that Free-AT and Free–TRADES also have a better generalization performance on the clean data than Vanilla-AT and TRADES. Our theoretical results suggest that the training process of free AT is algorithmically more stable than vanilla AT, therefore the free AT could be similarly expected to generalize better on clean data than vanilla AT. The numerical results

---

**Algorithm 4** Free–TRADES Adversarial Training Algorithm $A_{\text{Free–TRADES}}$

---

1: **Input:** Training samples $S$, perturbation set $\Delta$, step size of model weight $\alpha_w$, learning rate of attack $\alpha_\delta$, free step $m$, mini-batch size $b$, number of iterations $T$, TRADES coefficient $\lambda$
2: **for** step $\leftarrow 1, \cdots, T/m$ **do**
3:     Uniformly random mini-batch $B \subset S$ of size $b$
4:     $\delta := [\delta_j]_{\{j:x_j,y_j \in B\}} \leftarrow \text{Uniform}(\Delta^b)$
5:     **for** iteration $i \leftarrow 1, \cdots, m$ **do**
6:         Compute weight gradient and attack gradient by backpropagation:
7:         $g_w \leftarrow \frac{1}{b} \sum_{x_j,y_j \in B} \nabla_w \tilde{h}_\lambda(w, \delta_j; x_j, y_j)$, and $g_\delta \leftarrow [\nabla_\delta \tilde{h}_\lambda(w, \delta_j; x_j, y_j)]_{\{j:x_j,y_j \in B\}}$
8:         Update $w$ with mini-batch gradient descent: $w \leftarrow w - \alpha_w g_w$
9:         Update $\delta$ with projected gradient ascent: $\delta \leftarrow [\mathcal{P}_\Delta(\delta_j + \alpha_\delta \pi_\Delta(g_{\delta_j}))]_{\{j:x_j,y_j \in B\}}$
10:     **end for**
11: **end for**

---

Table 3: Robust generalization performance of the TRADES and Free-TRADES algorithms for ResNet18 models adversarially trained against $\mathcal{L}_2$ and $\mathcal{L}_\infty$ attacks on CIFAR-10 and CIFAR-100. We set TRADES coefficient $\lambda = 1/6$, free steps $m = 4$, and other details following from B.1. We run five independent trials and report the mean and standard deviation.

| Results (%) | CIFAR-10, $\mathcal{L}_2$ attack | | CIFAR-10, $\mathcal{L}_\infty$ attack | |
| --- | --- | --- | --- | --- |
| | TRADES | Free-TRADES | TRADES | Free-TRADES |
| Train Acc. | $99.1 \pm 0.1$ | $83.4 \pm 0.3$ | $85.6 \pm 0.3$ | $61.2 \pm 0.8$ |
| Test Acc. | $66.3 \pm 0.3$ | $68.2 \pm 0.2$ | $50.4 \pm 0.3$ | $49.8 \pm 0.5$ |
| Gen. Gap | $32.8 \pm 0.4$ | $15.2 \pm 0.2$ | $35.3 \pm 0.1$ | $11.4 \pm 0.3$ |

| Results (%) | CIFAR-100, $\mathcal{L}_2$ attack | | CIFAR-100, $\mathcal{L}_\infty$ attack | |
| --- | --- | --- | --- | --- |
| | TRADES | Free-TRADES | TRADES | Free-TRADES |
| Train Acc. | $99.6 \pm 0.1$ | $81.2 \pm 0.8$ | $83.5 \pm 0.7$ | $46.5 \pm 0.4$ |
| Test Acc. | $35.6 \pm 0.3$ | $40.0 \pm 0.2$ | $25.3 \pm 0.3$ | $25.4 \pm 0.3$ |
| Gen. Gap | $64.0 \pm 0.2$ | $41.1 \pm 0.9$ | $57.3 \pm 0.6$ | $21.1 \pm 0.4$ |

shown in Table 5 in Appendix B.1 are consistent with this intuition. The theoretical analysis of $A_{\text{Free–TRADES}}$ will be an interesting future direction for our work.

## 8   Conclusion

In this work, we studied the role of min-max optimization algorithms in the generalization performance of adversarial training methods. We focused on the widely-used free adversarial training method and, leveraging the algorithmic stability framework we compared its generalization behavior with that of vanilla adversarial training. Our generalization bounds suggest that not only can the free AT approach lead to a faster optimization compared to the vanilla AT, but also it leads to a lower generalization gap between the performance on training and test data. We note that our theoretical conclusions are based on the upper bounds following the algorithmic stability-based generalization analysis, and an interesting topic for future study is to prove a similar result for the actual generalization gap under simple linear or shallow neural net classifiers. Another future direction could be to extend our theoretical analysis of the simultaneous optimization updates to other adversarial training methods such as ALP (Kannan et al., 2018).

## Acknowledgment

The work of Farzan Farnia is partially supported by a grant from the Research Grants Council of the Hong Kong Special Administrative Region, China, Project 14209920, and is partially supported by a CUHK Direct Research Grant with CUHK Project No. 4055164.

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

## A Proof of Stability Generalization Bounds

We first prove the Lipschitzness of $\max_{\delta \in \Delta} h(w, \delta; x)$ by the following Lemma:

**Lemma 1.** *Define* $h_{max}(w; x) := \max_\delta h(w, \delta; x)$. *If* $h(w, \delta; x)$ *satisfies Assumption 1, then* $h_{max}$ *is* $L_w$-*Lipschitz in* $w$ *for any fixed* $x$.

*Proof.* For any $w, w'$ and $x \in X$, without loss of generality assume that $h_{\max}(w) \geq h_{\max}(w')$, then upon defining $\delta^* = \arg\max_{\delta \in \Delta} h(w, \delta; x)$ we have

$$
\begin{aligned}
|h_{\max}(w; x) - h_{\max}(w'; x)| &= h(w, \delta^*; x) - \max_{\delta \in \Delta} h(w', \delta; x) \\
&\leq h(w, \delta^*; x) - h(w', \delta^*; x) \\
&\leq L_w \|w - w'\|,
\end{aligned}
$$

thus completes the proof. $\qquad\square$

Another important observation is that similar to Lemma 3.11 in Hardt et al. (2015), mini-batch gradient descent typically makes several steps before it encounters the one example on which two datasets in stability analysis differ.

**Lemma 2.** *Suppose* $h_{max}(w; x) := \max_\delta h(w, \delta; x)$ *is bounded as* $h_{max} \in [0, 1]$. *By applying mini-batch gradient descent for two datasets* $S, S'$ *that only differ in only one sample* $s$, *denote by* $w_t$ *and* $w'_t$ *the output after* $t$ *steps respectively and define* $d_t^{(w)} := \|w_t - w'_t\|$. *Then for any* $x$ *and* $t_0$,

$$
\mathbb{E}[|h_{max}(w_t; x) - h_{max}(w'_t; x)|] \leq \frac{bt_0}{n} + L_w \mathbb{E}[d_t^{(w)} | d_{t_0}^{(w)} = 0].
$$

*Proof.* Denote the event $E = \mathbf{1}_{\{d_{t_0}^{(w)} = 0\}}$. Noting that if the sample $s$ is not visited before the $t_0$-th step, $w_t = w'_t$ since the updates are the same, hence

$$
\Pr(E^c) \leq \sum_{t=1}^{t_0} \Pr(s \in B_t) = \frac{bt_0}{n}.
$$

Then, by the law of total probability,

$$\mathbb{E}[|h_{\max}(w_t; x) - h_{\max}(w'_t; x)|] = \Pr(E)\mathbb{E}[|h_{\max}(w_t; x) - h_{\max}(w'_t; x)| \mid E]$$
$$+ \Pr(E^c)\mathbb{E}[|h_{\max}(w_t; x) - h_{\max}(w'_t; x)| \mid E^c]$$
$$\leq L_w \mathbb{E}[d_t^{(w)} | d_{t_0}^{(w)} = 0] + \frac{bt_0}{n},$$

where the last step comes from the Lipschitzness proved in Lemma 1. $\qquad\square$

Equipped with Lemmas 1 and 2, it remains to bound $\mathbb{E}[d_T^{(w)} | d_{t_0}^{(w)} = 0]$. The following lemma allows us to do this, given that for every $t$, $\mathbb{E}[d_t^{(w)}]$ is recursively bounded by the previous $\mathbb{E}[d_{t-1}^{(w)}]$.

**Lemma 3.** *Suppose that for every step $t$, the expected distance between weight parameters is bounded by the following recursion for some constants $\nu$ and $\xi$ (depending on the algorithm A):*

$$\mathbb{E}[d_t^{(w)}] \leq \left(1 + \frac{\nu}{t}\right) \mathbb{E}[d_{t-1}^{(w)}] + \frac{\nu}{nt}\xi. \tag{12}$$

*Then, after $T$ steps, the generalization adversarial risk can be bounded by*

$$\mathcal{E}_{gen}(A) \leq \frac{b}{n}\left(1 + \frac{1}{\nu}\right)\left(\frac{1}{b}L_w\xi\nu\right)^{\frac{1}{\nu+1}} T^{\frac{\nu}{\nu+1}}.$$

*Proof.* Conditioned on $d_{t_0}^{(w)} = 0$ for some $t_0$, by the recursion bound equation 12 we have

$$\mathbb{E}[d_T^{(w)} | d_{t_0}^{(w)} = 0] + \frac{\xi}{n} \leq \prod_{t=t_0+1}^{T}\left(1 + \frac{\nu}{t}\right) \cdot \frac{\xi}{n} \leq \prod_{t=t_0+1}^{T} \exp\left(\frac{\nu}{t}\right) \cdot \frac{\xi}{n} = \exp\left(\sum_{t=t_0+1}^{T} \frac{\nu}{t}\right) \cdot \frac{\xi}{n}$$
$$\leq \exp\left(\nu \log\left(\frac{T}{t_0}\right)\right) \cdot \frac{\xi}{n} = \frac{\xi}{n}\left(\frac{T}{t_0}\right)^{\nu}.$$

By Lemma 2, we further obtain

$$\mathbb{E}[|h_{\max}(w_t; x) - h_{\max}(w'_t; x)|] \leq L_w\frac{\xi}{n}\left(\frac{T}{t_0}\right)^{\nu} + \frac{bt_0}{n}.$$

The bound is maximized at

$$t_0 = \left(\frac{1}{b}L_w\xi T^{\nu}\nu\right)^{\frac{1}{\nu+1}}.$$

Combining this bound with Theorem 1 finally gives us

$$\mathcal{E}_{gen}(A) \leq \sup_{S,S',x} \mathbb{E}[|h_{\max}(w_t; x) - h_{\max}(w'_t; x)|] \leq \frac{b}{n}\left(1 + \frac{1}{\nu}\right)\left(\frac{1}{b}L_w\xi\nu\right)^{\frac{1}{\nu+1}} T^{\frac{\nu}{\nu+1}},$$

hence the proof is complete. $\qquad\square$

### A.1 Proof of Theorem 2

**Lemma 4** (Growth Lemma of $A_{\text{Vanilla}}$)**.** *Consider two datasets $S, S'$ differ in only one sample $s$. Then the following recursion holds for any step $t$*

$$\mathbb{E}[d_t^{(w)}] \leq (1 + \alpha_{w,t}\beta)\mathbb{E}[d_{t-1}^{(w)}] + \frac{\alpha_{w,t}\beta}{n}\left(2\varepsilon n + \frac{2L}{\beta}\right).$$

*Proof.* At step $t$, let $B_t, B_t'$ denote the mini-batches respectively. If $s \notin B_t$, we have

$$d_t^{(w)} = \left\| w_{t-1} - \frac{\alpha_{w,t}}{b} \sum_{x_j \in B_t} \nabla_w h(w_{t-1}, \delta_j; x_j) - w_{t-1}' + \frac{\alpha_{w,t}}{b} \sum_{x_j' \in B_t'} \nabla_w h(w_{t-1}', \delta_j'; x_j') \right\| \tag{13}$$

$$\leq \|w_{t-1} - w_{t-1}'\| + \frac{\alpha_{w,t}}{b} \sum_{x_j \in B_t} \left\| \nabla_w h(w_{t-1}, \delta_j; x_j) - \nabla_w h(w_{t-1}', \delta_j; x_j) \right\| \tag{14}$$

$$+ \frac{\alpha_{w,t}}{b} \sum_{x_j \in B_t} \left\| \nabla_w h(w_{t-1}', \delta_j; x_j) - \nabla_w h(w_{t-1}', \delta_j'; x_j) \right\| \tag{15}$$

$$\leq \|w_{t-1} - w_{t-1}'\| + \frac{\alpha_{w,t}}{b} \sum_{x_j \in B_t} \beta \|w_{t-1} - w_{t-1}'\| + \frac{\alpha_{w,t}}{b} \sum_{x_j \in B_t} \beta \left\| \delta_j - \delta_j' \right\| \tag{16}$$

$$= (1 + \alpha_{w,t}\beta) d_{t-1}^{(w)} + 2\varepsilon\alpha_{w,t}\beta, \tag{17}$$

where the last inequality is because the perturbation set $\Delta = \{\delta : \|\delta\| \leq \varepsilon\}$ hence $\left\| \delta_j - \delta_j' \right\| \leq 2\varepsilon$. If $s \in B_t$, by the Lipschitzness of $h$ we can bound $\|\nabla_w h(w, \delta_s; s)\| \leq L$ for all $w, \delta, s$. Hence

$$\left\| w_{t-1} - \alpha_{w,t} \nabla_w h(w_{t-1}, \delta_s; s) - w_{t-1}' + \alpha_{w,t} \nabla_w h(w_{t-1}', \delta_s'; s') \right\| \leq d_{t-1}^{(w)} + 2\alpha_{w,t} L.$$

Similar to equation 17 we can further bound

$$d_t^{(w)} = \left\| w_{t-1} - \frac{\alpha_{w,t}}{b} \sum_{x_j \in B_t} \nabla_w h(w_{t-1}, \delta_j; x_j) - w_{t-1}' + \frac{\alpha_{w,t}}{b} \sum_{x_j' \in B_t'} \nabla_w h(w_{t-1}', \delta_j'; x_j') \right\| \tag{18}$$

$$\leq \frac{b-1}{b} \left( (1 + \alpha_{w,t}\beta) d_{t-1}^{(w)} + 2\varepsilon\alpha_{w,t}\beta \right) + \frac{1}{b} \left( d_{t-1}^{(w)} + 2\alpha_{w,t} L \right). \tag{19}$$

Since $B_t$ is randomly drawn from $S$, $\Pr(s \in B_t) = \frac{b}{n}$. Combining equations 17 and 19, by the law of total probability we have

$$\mathbb{E}[d_t^{(w)}] \leq (1 + \alpha_{w,t}\beta)\mathbb{E}[d_{t-1}^{(w)}] + 2\varepsilon\alpha_{w,t}\beta + \frac{2}{n}\alpha_{w,t} L,$$

hence we finish the proof of Lemma 4. $\qquad\square$

By Lemma 4, upon plugging $\nu = \beta c$ and $\xi = 2\varepsilon n + \frac{2L}{\beta}$ into Lemma 3 we obtain the desired result

$$\mathcal{E}_{\text{gen}}(A_{\text{Vanilla}}) \leq \frac{b}{n} \left( 1 + \frac{1}{\beta c} \right) \left( \frac{2L_w c}{b}(\varepsilon\beta n + L) \right)^{\frac{1}{\beta c + 1}} T^{\frac{\beta c}{\beta c + 1}}.$$

## A.2 Proof of Theorem 4

**Lemma 5** (Iteration-wise Growth Lemma of $A_{\text{Free}}$). *Consider two datasets $S, S'$ differ in only one sample $s$. At iteration $i$ of step $t$, let $B_t, B_t'$ denote the mini-batches respectively, let $w_{t,i}, w_{t,i}'$ denote the model parameters and $\delta_{t,i}, \delta_{t,i}'$ denote the perturbations respectively, and let $d_{t,i}^{(w)} := \|w_{t,i} - w_{t,i}'\|$, $d_{t,i}^{(\delta)} := \frac{1}{b} \sum_{j:x_j \in B_t} \|(\delta_{t,i})_j - (\delta_{t,i}')_j\|$. Define the expansivity matrix*

$$\eta_t := \begin{bmatrix} 1 + \alpha_{w,t}\beta & \alpha_{w,t}\beta \\ \alpha_\delta\varepsilon\psi\beta & 1 + \alpha_\delta\varepsilon\psi\beta \end{bmatrix}. \tag{20}$$

*Then we have*

$$\begin{bmatrix} d_{t,i+1}^{(w)} \\ d_{t,i+1}^{(\delta)} \end{bmatrix} \leq \eta_t \cdot \begin{bmatrix} d_{t,i}^{(w)} \\ d_{t,i}^{(\delta)} \end{bmatrix} + \mathbf{1}_{\{s \in B_t\}} \begin{bmatrix} \frac{2}{b}\alpha_{w,t} L \\ \frac{2}{b}\alpha_\delta\varepsilon\psi L \end{bmatrix}.$$

*Proof.* If $s \notin B_t$, which implies $B_t = B_t'$, we have

$$d_{t,i+1}^{(w)} = \left\| w_{t,i} - \frac{\alpha_{w,t}}{b} \sum_{x_j \in B_t} \nabla_w h(w_{t,i}, (\delta_{t,i})_j; x_j) - w_{t,i}' + \frac{\alpha_{w,t}}{b} \sum_{x_j' \in B_t'} \nabla_w h(w_{t,i}', (\delta_{t,i}')_j; x_j') \right\| \tag{21}$$

$$\leq ||w_{t,i} - w_{t,i}'|| + \frac{\alpha_{w,t}}{b} \sum_{x_j \in B_t} ||\nabla_w h(w_{t,i}, (\delta_{t,i})_j; x_j) - \nabla_w h(w_{t,i}', (\delta_{t,i})_j; x_j)|| \tag{22}$$

$$+ \frac{\alpha_{w,t}}{b} \sum_{x_j \in B_t} ||\nabla_w h(w_{t,i}', (\delta_{t,i})_j; x_j) - \nabla_w h(w_{t,i}', (\delta_{t,i}')_j; x_j)|| \tag{23}$$

$$\leq ||w_{t,i} - w_{t,i}'|| + \frac{\alpha_{w,t}}{b} \sum_{x_j \in B_t} \beta ||w_{t,i} - w_{t,i}'|| + \frac{\alpha_{w,t}}{b} \sum_{x_j \in B_t} \beta \left\| (\delta_{t,i})_j - (\delta_{t,i}')_j \right\| \tag{24}$$

$$= (1 + \alpha_{w,t}\beta) d_{t,i}^{(w)} + \alpha_{w,t}\beta d_{t,i}^{(\delta)}. \tag{25}$$

If $s \in B_t$, the gradient difference with respect to $s$ and $s'$ shall be separately bounded. By the Lipschitzness of $h$, we can bound the expansive property of the minimization step with respect to $s$ and $s'$ by

$$||w_{t,i} - \alpha_{w,t}\nabla_w h(w_{t,i}, (\delta_{t,i})_j; s) - w_{t,i}' + \alpha_{w,t}\nabla_w h(w_{t,i}', (\delta_{t,i}')_j; s')|| \leq d_{t,i}^{(w)} + 2\alpha_{w,t}L.$$

Similar to equation 25 we can further derive the bound for mini-batch gradient descent by

$$d_{t,i+1}^{(w)} = \left\| w_{t,i} - \frac{\alpha_{w,t}}{b} \sum_{x_j \in B_t} \nabla_w h(w_{t,i}, (\delta_{t,i})_j; x_j) - w_{t,i}' + \frac{\alpha_{w,t}}{b} \sum_{x_j \in B_t'} \nabla_w h(w_{t,i}', (\delta_{t,i}')_j; x_j') \right\|$$

$$\leq \frac{b-1}{b} \left( (1 + \alpha_{w,t}\beta) d_{t,i}^{(w)} + \alpha_{w,t}\beta d_{t,i}^{(\delta)} \right) + \frac{1}{b} \left( d_{t,i}^{(w)} + 2\alpha_{w,t}L \right)$$

$$\leq (1 + \alpha_{w,t}\beta) d_{t,i}^{(w)} + \alpha_{w,t}\beta d_{t,i}^{(\delta)} + \frac{2}{b}\alpha_{w,t}L.$$

We then proceed to bound $d_{t,i}^{(\delta)}$ recursively. When $\Delta = \{\delta : ||\delta|| \leq \varepsilon\}$, by the definition of projected gradient in equation 1, we have

$$\pi_\Delta(g) = \underset{\tilde{\delta} \in \text{ExtremePoints}(\Delta)}{\arg\min} ||g - \tilde{\delta}||^2 = \frac{\varepsilon g}{||g||}.$$

Since we assume that with probability 1 the norm of gradient $\nabla_\delta h(w, \delta; x)$ is lower bounded by $1/\psi$ for some constant $\psi > 0$, we can translate $\pi_\Delta$ into the projection onto the convex set $\Delta$. For any vector $g$ such that $||g|| \geq 1/\psi$,

$$\pi_\Delta(g) = \frac{\varepsilon g}{||g||} = \frac{\varepsilon \psi g}{\psi ||g||} = \frac{\varepsilon \psi g}{\max\{1, \varepsilon \psi ||g||/\varepsilon\}} = \underset{\delta \in \Delta}{\arg\min} ||\varepsilon \psi g - \delta||^2 = \mathcal{P}_\Delta(\varepsilon \psi g). \tag{26}$$

Since $\Delta$ is convex, the projection $\mathcal{P}_\Delta(\cdot)$ is 1-Lipschitz. So for all $j$ such that $x_j \in B$ we have

$$||(\delta_{t,i+1})_j - (\delta_{t,i+1}')_j|| = ||\mathcal{P}_\Delta((\delta_{t,i})_j + \alpha_\delta \pi_\Delta(g_{\delta_j})) - \mathcal{P}_\Delta((\delta_{t,i}')_j + \alpha_\delta \pi_\Delta(g_{\delta_j}'))||$$

$$\leq ||(\delta_{t,i})_j + \alpha_\delta \pi_\Delta(g_{\delta_j}) - (\delta_{t,i}')_j - \alpha_\delta \pi_\Delta(g_{\delta_j}')||$$

$$\leq ||(\delta_{t,i})_j - (\delta_{t,i}')_j|| + \alpha_\delta ||\mathcal{P}_\Delta(\varepsilon \psi g_{\delta_j}) - \mathcal{P}_\Delta(\varepsilon \psi g_{\delta_j}')||$$

$$\leq ||(\delta_{t,i})_j - (\delta_{t,i}')_j|| + \alpha_\delta \varepsilon \psi ||\nabla_\delta h(w_{t,i}, (\delta_{t,i})_j; x_j) - \nabla_\delta h(w_{t,i}', (\delta_{t,i}')_j; x_j')||.$$

If $x_j = x_j'$, by the smoothness we can further bound

$$||(\delta_{t,i+1})_j - (\delta_{t,i+1}')_j|| \leq ||(\delta_{t,i})_j - (\delta_{t,i}')_j|| + \alpha_\delta \varepsilon \psi ||\nabla_\delta h(w_{t,i}, (\delta_{t,i})_j; x_j) - \nabla_\delta h(w_{t,i}', (\delta_{t,i})_j; x_j)|| \tag{27}$$

$$+ \alpha_\delta \varepsilon \psi ||\nabla_\delta h(w_{t,i}', (\delta_{t,i})_j; x_j) - \nabla_\delta h(w_{t,i}', (\delta_{t,i}')_j; x_j)|| \tag{28}$$

$$\leq (1 + \alpha_\delta \varepsilon \psi \beta) ||(\delta_{t,i})_j - (\delta_{t,i}')_j|| + \alpha_\delta \varepsilon \psi \beta d_{t,i}^{(w)}. \tag{29}$$

Otherwise if $x_j = s \neq x'_j$, we can bound it by the Lipschitzness

$$||(\delta_{t,i+1})_j - (\delta'_{t,i+1})_j|| \leq ||(\delta_{t,i})_j - (\delta'_{t,i})_j|| + 2\alpha_\delta \varepsilon \psi L. \tag{30}$$

Upon combining equations 29 and 30, we obtain the desired recursion bound for $d_{t,i}^{(\delta)}$

$$d_{t,i+1}^{(\delta)} \leq (1 + \alpha_\delta \varepsilon \psi \beta) d_{t,i}^{(\delta)} + \alpha_\delta \varepsilon \psi \beta d_{t,i}^{(w)} + \mathbf{1}_{\{s \in B_t\}} \cdot \frac{2}{b} \alpha_\delta \varepsilon \psi L.$$

Finally, combining the above completes the proof. □

**Lemma 6** (Step-wise Growth Lemma of $A_{\text{Free}}$). *Consider two datasets $S, S'$ differ in only one sample $s$. Let $B_t, B'_t$ denote the mini-batches at step $t$ respectively, and let $d_t^{(w)} := ||w_{t,m} - w'_{t,m}||$ be the distance between weight parameters after step $t$. Over the randomness of $B_t$, we have*

$$\mathbb{E}[d_t^{(w)}] + \frac{2L}{n\beta} \leq \left(1 + \frac{\beta c}{t} \cdot (1 + \alpha_{w,t}\beta + \alpha_\delta \varepsilon \psi \beta)^{m-1}\right) \left(\mathbb{E}[d_{t-1}^{(w)}] + \frac{2L}{n\beta}\right).$$

*Proof.* Noting that by Lemma 5 we can obtain

$$\begin{bmatrix} d_{t,i+1}^{(w)} \\ d_{t,i+1}^{(\delta)} \end{bmatrix} + \mathbf{1}_{\{s \in B_t\}} \begin{bmatrix} 2L/b\beta \\ 0 \end{bmatrix} \leq \eta_t \cdot \begin{bmatrix} d_{t,i}^{(w)} \\ d_{t,i}^{(\delta)} \end{bmatrix} + \mathbf{1}_{\{s \in B_t\}} \begin{bmatrix} \frac{2L}{b\beta} + \frac{2}{b}\alpha_{w,t}L \\ \frac{2}{b}\alpha_\delta \varepsilon \psi L \end{bmatrix}$$

$$= \eta_t \cdot \left( \begin{bmatrix} d_{t,i}^{(w)} \\ d_{t,i}^{(\delta)} \end{bmatrix} + \mathbf{1}_{\{s \in B_t\}} \begin{bmatrix} 2L/b\beta \\ 0 \end{bmatrix} \right).$$

Since the updates are repeated for $m$ iterations in one step, by induction we have

$$\begin{bmatrix} d_{t,m}^{(w)} \\ d_{t,m}^{(\delta)} \end{bmatrix} + \mathbf{1}_{\{s \in B_t\}} \begin{bmatrix} 2L/b\beta \\ 0 \end{bmatrix} \leq \eta_t^m \cdot \left( \begin{bmatrix} d_{t,0}^{(w)} \\ 0 \end{bmatrix} + \mathbf{1}_{\{s \in B_t\}} \begin{bmatrix} 2L/b\beta \\ 0 \end{bmatrix} \right).$$

Denote $\alpha := \alpha_{w,t}\beta$ and $r := \alpha_\delta \varepsilon \psi / \alpha_{w,t}$ for simplicity. By the definition in equation 20, we can calculate the eigendecomposition of $\eta_t$

$$\eta_t = \begin{bmatrix} 1+\alpha & \alpha \\ \alpha r & 1+\alpha r \end{bmatrix} = \begin{bmatrix} \frac{1}{r} & -1 \\ 1 & 1 \end{bmatrix} \begin{bmatrix} 1+\alpha(r+1) & 0 \\ 0 & 1 \end{bmatrix} \begin{bmatrix} \frac{r}{r+1} & \frac{r}{r+1} \\ -\frac{r}{r+1} & \frac{1}{r+1} \end{bmatrix}.$$

Denoting $d_0 := d_{t,0}^{(w)} + \mathbf{1}_{\{s \in B_t\}} \frac{2L}{b\beta}$ and $d_m := d_{t,m}^{(w)} + \mathbf{1}_{\{s \in B_t\}} \frac{2L}{b\beta}$, we can solve the recursion

$$\begin{bmatrix} d_m \\ d_{t,m}^{(\delta)} \end{bmatrix} \leq \eta_t^m \cdot \begin{bmatrix} d_0 \\ 0 \end{bmatrix} = \begin{bmatrix} \frac{1}{r} & -1 \\ 1 & 1 \end{bmatrix} \begin{bmatrix} (1+\alpha(r+1))^m & 0 \\ 0 & 1 \end{bmatrix} \begin{bmatrix} \frac{r}{r+1} & \frac{r}{r+1} \\ -\frac{r}{r+1} & \frac{1}{r+1} \end{bmatrix} \begin{bmatrix} d_0 \\ 0 \end{bmatrix}$$

$$= \begin{bmatrix} \frac{1}{r} & -1 \\ 1 & 1 \end{bmatrix} \begin{bmatrix} (1+\alpha(r+1))^m & 0 \\ 0 & 1 \end{bmatrix} \begin{bmatrix} \frac{r}{r+1}d_0 \\ -\frac{r}{r+1}d_0 \end{bmatrix}$$

$$= \begin{bmatrix} \frac{1}{r} & -1 \\ 1 & 1 \end{bmatrix} \begin{bmatrix} (1+\alpha(r+1))^m \frac{r}{r+1}d_0 \\ -\frac{r}{r+1}d_0 \end{bmatrix}$$

$$= \begin{bmatrix} \frac{r+(1+\alpha(r+1))^m}{r+1}d_0 \\ \frac{r((1+\alpha(r+1))^m-1)}{r+1}d_0 \end{bmatrix}.$$

Noting that by assumption $\alpha_{w,t} \leq c/mt$, upon plugging in $\alpha = \alpha_{w,t}\beta$ and $r = \alpha_\delta \varepsilon \psi / \alpha_{w,t}$,

$$\frac{r+(1+\alpha(r+1))^m}{r+1} = 1 + \frac{(1+\alpha(r+1))^m - 1}{r+1}$$

$$= 1 + \alpha \sum_{j=0}^{m-1} (1+\alpha(r+1))^j$$

$$\leq 1 + \alpha \cdot m(1+\alpha(r+1))^{m-1}$$

$$\leq 1 + \frac{\beta c}{t}(1 + \alpha_{w,t}\beta + \alpha_\delta \varepsilon \psi \beta)^{m-1}.$$

Since $d_t^{(w)} = d_{t,m}^{(w)}$ and $d_{t-1}^{(w)} = d_{t,0}^{(w)}$, we obtain that

$$d_t^{(w)} + \mathbf{1}_{\{s \in B_t\}} \frac{2L}{b\beta} \le \left(1 + \frac{\beta c}{t} \cdot (1 + \alpha_{w,t}\beta + \alpha_\delta \varepsilon \psi \beta)^{m-1}\right)\left(d_{t-1}^{(w)} + \mathbf{1}_{\{s \in B_t\}} \frac{2L}{b\beta}\right).$$

Since $B_t$ is drawn uniformly randomly from $S$, $\Pr(s \in B_t) = \frac{b}{n}$. By the law of total probability, we complete the proof. $\qquad\square$

By Lemma 6, since $\alpha_{w,t} \le c/mt \le c/m$, upon plugging $\nu = \beta c(1 + \beta c/m + \alpha_\delta \varepsilon \psi \beta)^{m-1} = \lambda_{\text{Free}}$ and $\xi = \frac{2L}{\beta}$ into Lemma 3 we obtain that after $T/m$ steps,

$$\mathcal{E}_{\text{gen}}(A_{\text{Vanilla}}) \le \frac{b}{n}\left(1 + \frac{1}{\lambda_{\text{Free}}}\right)\left(\frac{2LL_w}{b\beta}\lambda_{\text{Free}}\right)^{\frac{1}{\lambda_{\text{Free}}+1}}\left(\frac{T}{m}\right)^{\frac{\lambda_{\text{Free}}}{\lambda_{\text{Free}}+1}}.$$

### A.3 Proof of Theorem 5

To prove Theorem 5, we start with the following growth lemma of $A_{\text{Fast}}$.

**Lemma 7** (Growth Lemma of $A_{\text{Fast}}$). *Consider two datasets $S, S'$ differ in only one sample $s$. Then the following recursion holds for any step $t$*

$$\mathbb{E}[d_t^{(w)}] \le (1 + \alpha_{w,t}\beta(1 + \tilde\alpha_\delta \varepsilon \psi \beta))\mathbb{E}[d_{t-1}^{(w)}] + \frac{2}{n}\alpha_{w,t}L.$$

*Proof.* We first bound the difference between $\delta_j$ and $\delta'_j$. At step $t$, let $B_t, B'_t$ denote the mini-batches respectively. By equation 26, we have $\pi_\Delta(g) = \mathcal{P}_\Delta(\varepsilon \psi g)$ if $\|g\| \ge 1/\psi$. Since $\Delta$ is convex, the projection $\mathcal{P}_\Delta(\cdot)$ is 1-Lipschitz. So for all $j$ such that $x_j \in B$ we have

$$\begin{aligned}
\left\|\delta_j - \delta'_j\right\| &= \left\|\mathcal{P}_\Delta(\tilde\delta_j + \tilde\alpha_\delta \pi_\Delta(g_{\delta_j})) - \mathcal{P}_\Delta(\tilde\delta_j + \tilde\alpha_\delta \pi_\Delta(g_{\delta'_j}))\right\| \\
&\le \left\|(\tilde\delta_j + \tilde\alpha_\delta \pi_\Delta(g_{\delta_j})) - (\tilde\delta_j + \tilde\alpha_\delta \pi_\Delta(g_{\delta'_j}))\right\| \\
&\le \tilde\alpha_\delta \left\|\mathcal{P}_\Delta(\varepsilon \psi g_{\delta_j}) - \mathcal{P}_\Delta(\varepsilon \psi g_{\delta'_j})\right\| \\
&\le \tilde\alpha_\delta \varepsilon \psi \left\|\nabla_\delta h(w_{t-1}, \tilde\delta_j; x_j) - \nabla_\delta h(w'_{t-1}, \tilde\delta_j; x'_j)\right\|
\end{aligned}$$

If $x_j = x'_j$, by smoothness we obtain $\|\delta_j - \delta'_j\| \le \tilde\alpha_\delta \varepsilon \psi \beta d_{t-1}^{(w)}$, so

$$\begin{aligned}
&\|\nabla_w h(w_{t-1}, \delta_j; x_j) - \nabla_w h(w'_{t-1}, \delta'_j; x'_j)\| \\
&\le \|\nabla_w h(w_{t-1}, \delta_j; x_j) - \nabla_w h(w'_{t-1}, \delta_j; x'_j)\| + \|\nabla_w h(w'_{t-1}, \delta_j; x_j) - \nabla_w h(w'_{t-1}, \delta'_j; x'_j)\| \\
&\le \beta\|w_{t-1} - w'_{t-1}\| + \beta\|\delta_j - \delta'_j\| \\
&\le \beta(1 + \tilde\alpha_\delta \varepsilon \psi \beta)d_{t-1}^{(w)}.
\end{aligned}$$

Otherwise if $x_j = s \ne x'_j$, we can only bound this term by the Lipschitzness

$$\|\nabla_w h(w_{t-1}, \delta_j; x_j) - \nabla_w h(w'_{t-1}, \delta'_j; x'_j)\| \le 2L.$$

Therefore, we can bound $d_t^{(w)}$ by the following recursion

$$\begin{aligned}
d_t^{(w)} &= \left\|w_{t-1} - \frac{\alpha_{w,t}}{b}\sum_{x_j \in B_t}\nabla_w h(w_{t-1}, \delta_j; x_j) - w'_{t-1} + \frac{\alpha_{w,t}}{b}\sum_{x'_j \in B'_t}\nabla_w h(w'_{t-1}, \delta'_j; x'_j)\right\| \\
&\le \|w_{t-1} - w'_{t-1}\| + \frac{\alpha_{w,t}}{b}\sum_{x_j \in B_t}\left\|\nabla_w h(w_{t-1}, \delta_j; x_j) - \nabla_w h(w'_{t-1}, \delta'_j; x'_j)\right\| \\
&\le d_{t-1}^{(w)} + \alpha_{w,t}\beta(1 + \tilde\alpha_\delta \varepsilon \psi \beta)d_{t-1}^{(w)} + \mathbf{1}_{\{s \in B_t\}} \cdot \frac{2\alpha_{w,t}L}{b}
\end{aligned}$$

Since $B_t$ is randomly drawn from $S$, $\Pr(s \in B_t) = \frac{b}{n}$. So by the law of total probability we have

$$\mathbb{E}[d_t^{(w)}] \leq (1 + \alpha_{w,t}\beta(1 + \tilde{\alpha}_\delta\varepsilon\psi\beta))\mathbb{E}[d_{t-1}^{(w)}] + \frac{2}{n}\alpha_{w,t}L,$$

hence we finish the proof of Lemma 7. □

Armed with Lemmas 3 and 7, by letting $\nu = \beta c(1 + \tilde{\alpha}_\delta\varepsilon\psi\beta)$ and $\xi = \frac{2L}{\beta(1+\tilde{\alpha}_\delta\varepsilon\psi\beta)}$ we obtain

$$\mathcal{E}_{\text{gen}}(A_{\text{Fast}}) \leq \frac{b}{n}\left(1 + \frac{1}{\beta c(1 + \tilde{\alpha}_\delta\varepsilon\psi\beta)}\right)\left(\frac{2cLL_w}{b}\right)^{\frac{1}{\beta c(1+\tilde{\alpha}_\delta\varepsilon\psi\beta)+1}}T^{\frac{\beta c(1+\tilde{\alpha}_\delta\varepsilon\psi\beta)}{\beta c(1+\tilde{\alpha}_\delta\varepsilon\psi\beta)+1}},$$

thus complete the proof.

# B  Additional Numerical Results

## B.1  Robust Overfitting During Training Process

**Implementation Details:**  For the training process of networks, we follow the standards in the literature (Madry et al., 2017; Rice et al., 2020). We apply mini-batch gradient descent with batch size $b = 128$. Weight decay is set to be $2 \times 10^{-4}$. We adopt a piecewise learning rate decay schedule, starting with 0.1 and decaying by a factor of 10 at the 100th and 150th epochs, for 200 total epochs. For the vanilla algorithm, we apply PGD adversaries with 1, 7, or 10 iterations and set the step size as $\varepsilon$, $\varepsilon/4$, or $\varepsilon/4$ respectively. For the free algorithm, since it repeats $m$ iterations at each step, we use $200/m$ epochs to match their training iterations for fair comparison. For the adversaries, we consider both $\mathcal{L}_2$-norm attack of radius $\varepsilon = 128/255$, and $\mathcal{L}_\infty$-norm attack of radius $\varepsilon = 8/255$ (except for Tiny-ImageNet $\varepsilon = 4/255$).

We extend the experiments in Section 6 using different free steps $m$, various neural network architectures, and other datasets. We use ResNet18 for CIFAR-10 and CIFAR-100, ResNet50 for Tiny-ImageNet, and VGG19 for SVHN. We apply both $\mathcal{L}_2$ and $\mathcal{L}_\infty$ adversaries and provide the plots of training curves for CIFAR-10 and CIFAR-100 in Figure 5. We can observe that the vanilla training suffers from robust overfitting, while the free algorithm with moderate free steps generalizes better and achieves comparable robustness to the vanilla algorithm.

Table 4: Robust training accuracy, testing accuracy, and generalization gap of the vanilla, fast, and free algorithms across Tiny-ImageNet and SVHN datasets.

| Dataset | Attack | Results (%) | Vanilla-7 | Vanilla-10 | Fast | Free-2 | Free-4 | Free-6 | Free-8 | Free-10 |
|---|---|---|---|---|---|---|---|---|---|---|
| Tiny-ImageNet | $\mathcal{L}_2$ | Train Acc. | 100.0 | 100.0 | 100.0 | 100.0 | 100.0 | 99.3 | 65.1 | 46.8 |
| | | Test Acc. | 22.9 | 23.0 | 23.5 | 23.8 | 24.8 | 22.7 | 23.6 | 23.8 |
| | | Gen. Gap | 77.1 | 77.0 | 76.5 | 76.2 | 75.2 | 76.6 | 41.5 | 23.0 |
| | $\mathcal{L}_\infty$ | Train Acc. | 100.0 | 100.0 | 100.0 | 99.2 | 99.8 | 96.2 | 60.2 | 39.5 |
| | | Test Acc. | 13.8 | 13.9 | 13.5 | 12.3 | 14.5 | 14.4 | 15.7 | 16.8 |
| | | Gen. Gap | 86.2 | 86.1 | 86.5 | 86.9 | 85.3 | 81.8 | 44.5 | 22.7 |
| SVHN | $\mathcal{L}_2$ | Train Acc. | 100.0 | 100.0 | 96.9 | 89.2 | 95.1 | 90.0 | 93.5 | 91.0 |
| | | Test Acc. | 61.4 | 61.2 | 60.8 | 61.6 | 60.8 | 62.3 | 61.6 | 62.5 |
| | | Gen. Gap | 38.6 | 38.8 | 36.1 | 27.6 | 34.3 | 27.7 | 31.9 | 28.5 |
| | $\mathcal{L}_\infty$ | Train Acc. | 89.1 | 88.9 | 55.1 | 49.6 | 55.4 | 63.8 | 56.7 | 56.9 |
| | | Test Acc. | 38.7 | 38.8 | 39.5 | 38.3 | 42.7 | 47.1 | 46.7 | 45.0 |
| | | Gen. Gap | 50.4 | 50.1 | 15.6 | 11.3 | 12.7 | 16.7 | 10.0 | 11.9 |

Table 5: Clean and robust training accuracy, testing accuracy, and generalization gap of the vanilla, free, TRADES, and Free–TRADES algorithms on CIFAR-10 dataset.

| Attack | Results (%) | Vanilla | Free | TRADES | Free–TRADES |
|---|---|---|---|---|---|
| $\mathcal{L}_2$ | Clean Train Acc. | 100.0 | 98.7 | 99.8 | 95.8 |
| | Clean Test Acc. | 88.1 | 88.6 | 86.2 | 86.8 |
| | Clean Gen. Gap | 11.9 | 10.1 | 13.6 | 9.0 |
| | Robust Train Acc. | 100.0 | 86.4 | 99.2 | 83.1 |
| | Robust Test Acc. | 65.6 | 66.0 | 66.2 | 68.0 |
| | Robust Gen. Gap | 34.4 | 20.4 | 33.0 | 15.1 |
| $\mathcal{L}_\infty$ | Clean Train Acc. | 99.9 | 95.1 | 97.7 | 91.0 |
| | Clean Test Acc. | 84.7 | 85.9 | 82.4 | 83.1 |
| | Clean Gen. Gap | 15.2 | 9.2 | 15.3 | 7.9 |
| | Robust Train Acc. | 95.1 | 63.7 | 85.5 | 60.2 |
| | Robust Test Acc. | 43.7 | 46.3 | 50.1 | 48.7 |
| | Robust Gen. Gap | 51.4 | 17.4 | 35.4 | 11.5 |

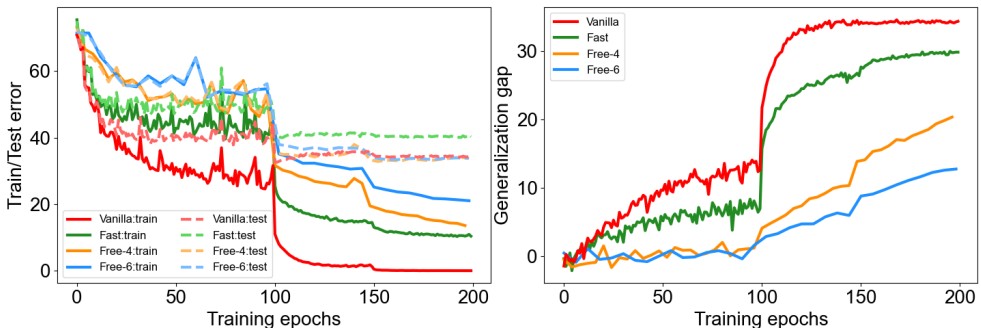

(a) Robust train/test error and generalization gap against $\mathcal{L}_2$-norm attack on CIFAR-10.

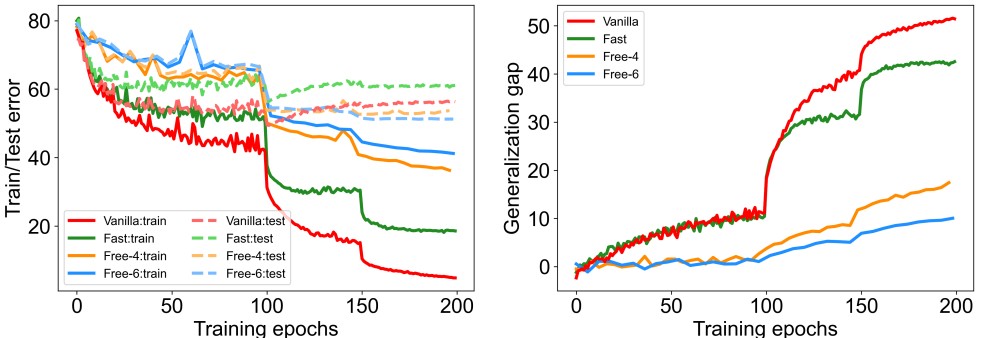

(b) Robust train/test error and generalization gap against $\mathcal{L}_\infty$-norm attack on CIFAR-10.

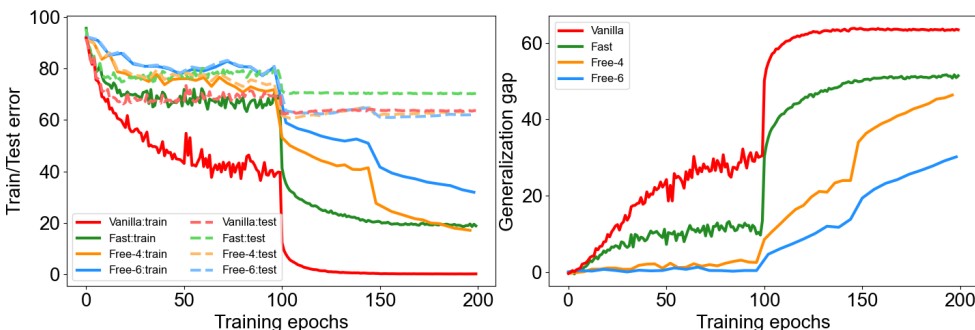

(c) Robust train/test error and generalization gap against $\mathcal{L}_2$-norm attack on CIFAR-100.

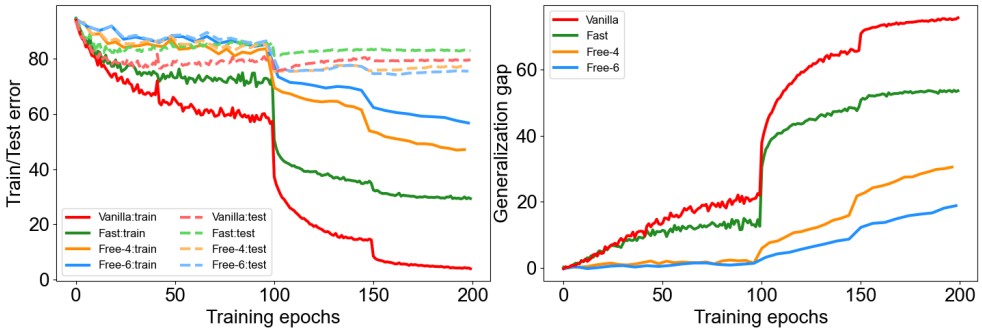

(d) Robust train/test error and generalization gap against $\mathcal{L}_\infty$-norm attack on CIFAR-100.

Figure 5: Learning curves of different algorithms for a ResNet18 model adversarially trained against $\mathcal{L}_2$-norm and $\mathcal{L}_\infty$-norm attacks on CIFAR-10 and CIFAR-100. The free curves are scaled horizontally by a factor of $m$ for clear comparison.

## B.2 Robustness Against Black-box Attack

We perform additional experiments to test the robust accuracy of models trained by vanilla, fast, and free algorithms against black-box attacks. In Figure 6, we demonstrate their accuracy against $\mathcal{L}_2$ and $\mathcal{L}_\infty$ square attacks with 5000 queries of various radius. In Figure 7, we use transferred attacks designed for other independently trained robust models.

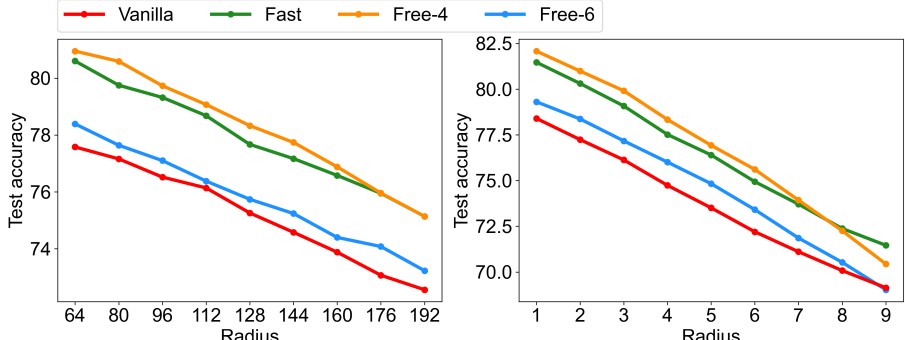

(a) Accuracy of adversarially trained ResNet18 models against $\mathcal{L}_2$ and $\mathcal{L}_\infty$ square attacks on CIFAR-10.

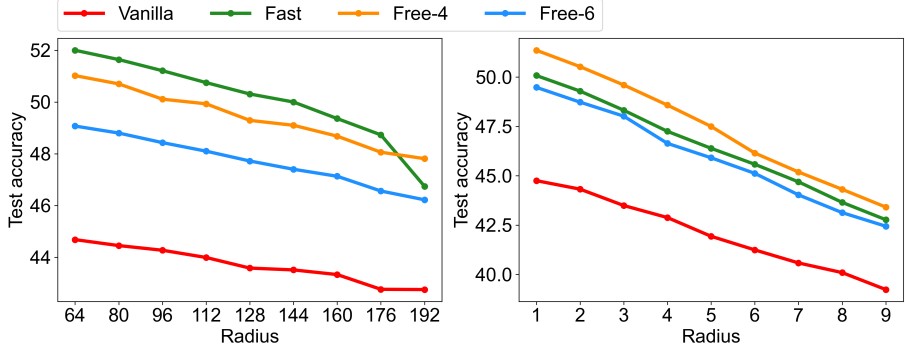

(b) Accuracy of adversarially trained ResNet18 models against $\mathcal{L}_2$ and $\mathcal{L}_\infty$ square attacks on CIFAR-100.

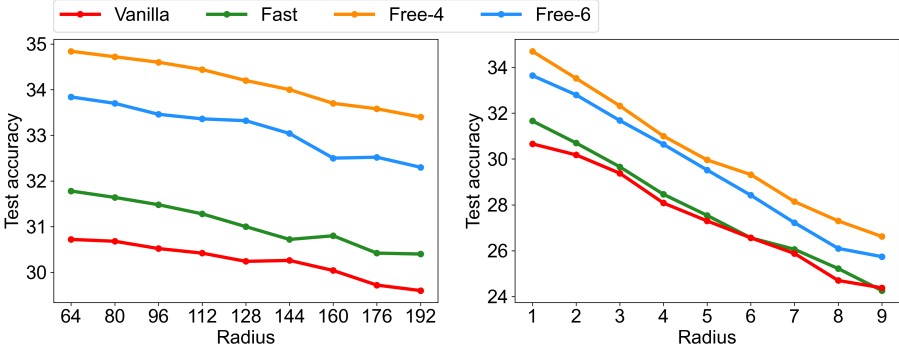

(c) Accuracy of adversarially trained ResNet50 models against $\mathcal{L}_2$ and $\mathcal{L}_\infty$ square attacks on Tiny-ImageNet.

Figure 6: Robust accuracy of adversarially trained models by vanilla, fast, and free algorithms against square attacks on CIFAR-10, CIFAR-100, and Tiny-ImageNet. The left figure applies $\mathcal{L}_2$ attacks of radius from 64 to 192, and the right figure applies $\mathcal{L}_\infty$ attacks of radius from 1 to 9.

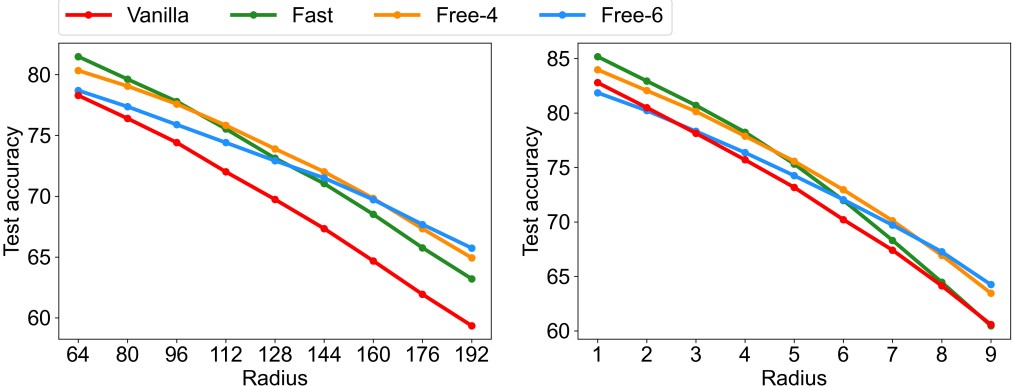

(a) Accuracy of adversarially trained ResNet18 models against $\mathcal{L}_2$ and $\mathcal{L}_\infty$ transferred attacks on CIFAR-10.

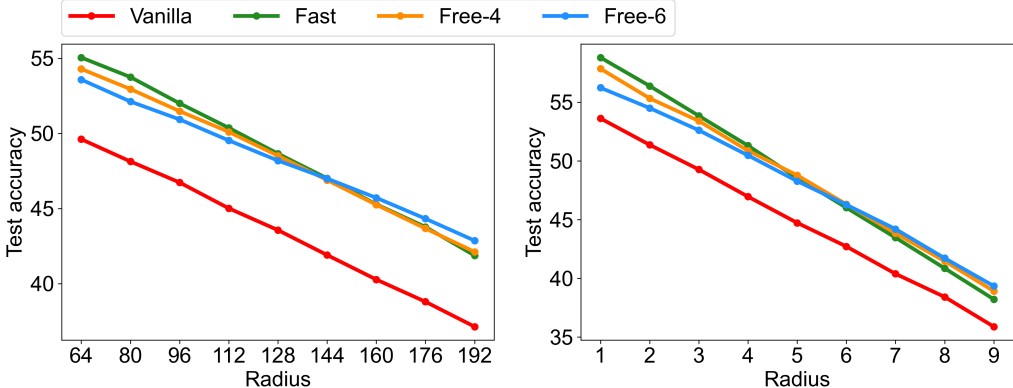

(b) Accuracy of adversarially trained ResNet18 models against $\mathcal{L}_2$ and $\mathcal{L}_\infty$ transferred attacks on CIFAR-100.

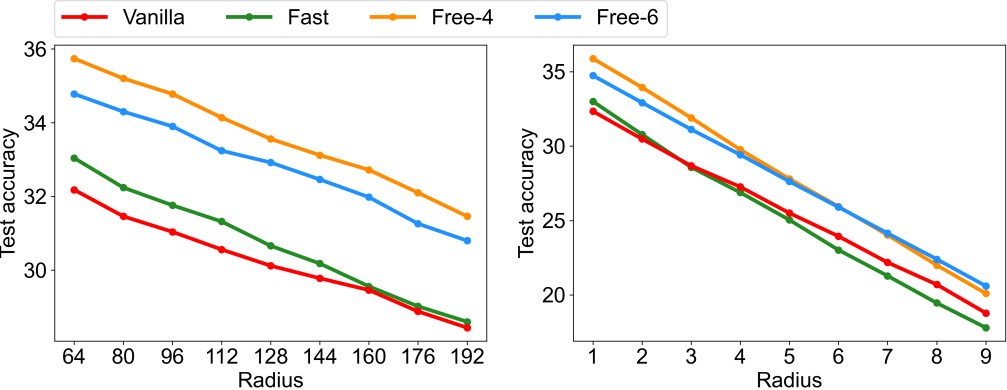

(c) Accuracy of adversarially trained ResNet50 models against $\mathcal{L}_2$ and $\mathcal{L}_\infty$ transferred attacks on Tiny-ImageNet.

Figure 7: Robust accuracy of models adversarially trained by vanilla, fast, and free algorithms against transferred attacks designed for other independently trained robust models on CIFAR-10, CIFAR-100, and Tiny-ImageNet. The left figure applies $\mathcal{L}_2$ attacks of radius ranging from 64 to 192, and the right figure applies $\mathcal{L}_\infty$ attacks of radius ranging from 1 to 9.

### B.3 Transferability

We further investigated the transferability of the adversarial examples designed for the models trained by the mentioned adversarial training algorithms. We computed the adversarial perturbations designed for the robust models and used them to attack other standard ERM-trained models. We test the transferability of models trained by vanilla, fast, and free algorithms. In Figure 8, we transfer the attacks to a standard ResNet18 model on CIFAR-100 and a standard ResNet50 model on Tiny-ImageNet. In Figure 9, we transfer the attacks to various standard models including ResNet18, ResNet50, and Wide-ResNet34 (Zagoruyko & Komodakis, 2016) on CIFAR-10.

Our numerical results suggest that the better generalization performance of the free algorithm could result in more transferable adversarial perturbations, which could be more detrimental to the performance of other unseen neural network models.

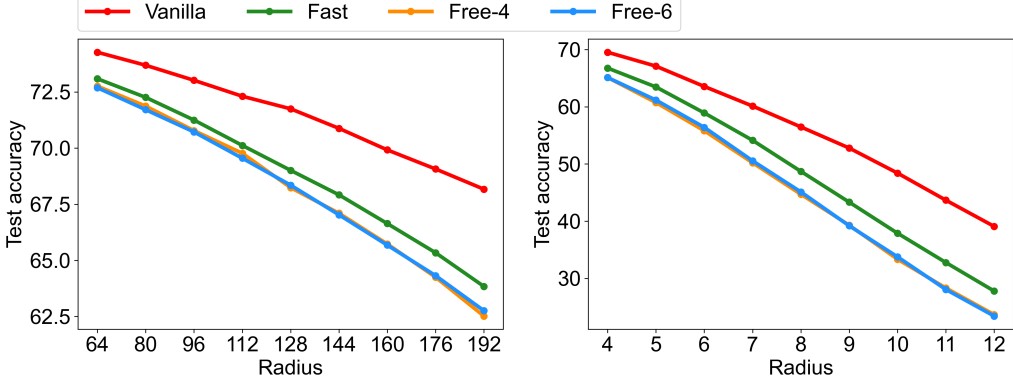

(a) Test accuracy of a standard ResNet18 target model against transferred attacks from adversarially trained models on CIFAR-100.

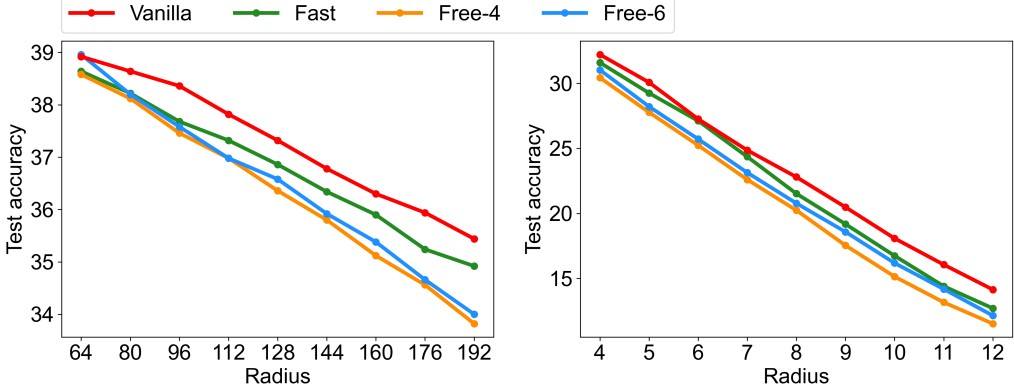

(b) Test accuracy of a standard ResNet50 target model against transferred attacks from adversarially trained models on Tiny-ImageNet.

Figure 8: Test accuracy of standard ResNet18 and ResNet50 target models against transferred attacks from models adversarially trained by vanilla, fast, and free algorithms on CIFAR-100 and Tiny-ImageNet. The left figure applies $\mathcal{L}_2$ attacks of radius ranging from 64 to 192, and the right figure applies $\mathcal{L}_\infty$ attacks of radius ranging from 4 to 12.

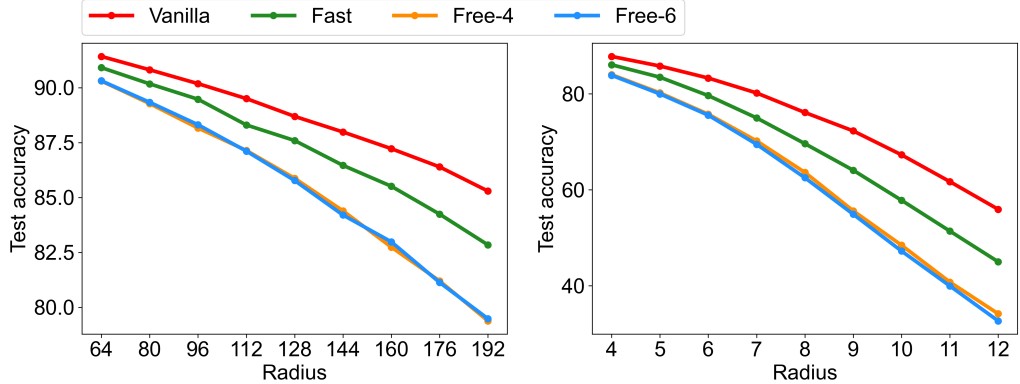

(a) Test accuracy of a standard ResNet18 target model against transferred attacks from adversarially trained ResNet18 models.

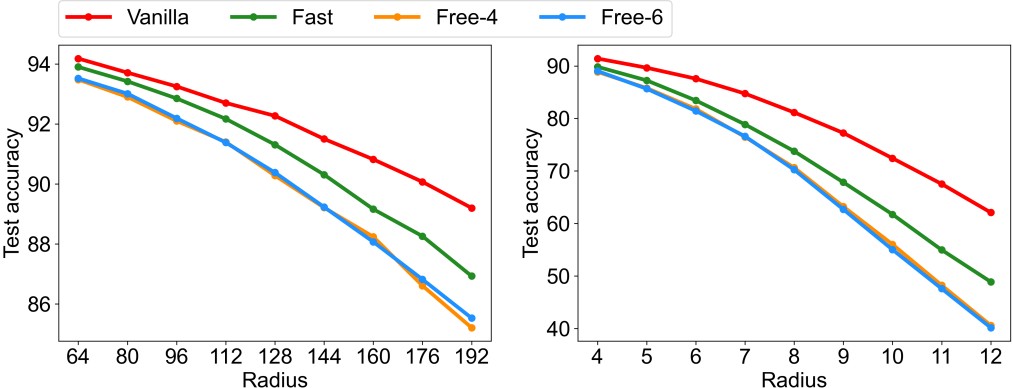

(b) Test accuracy of a standard ResNet50 target model against transferred attacks from adversarially trained ResNet18 models.

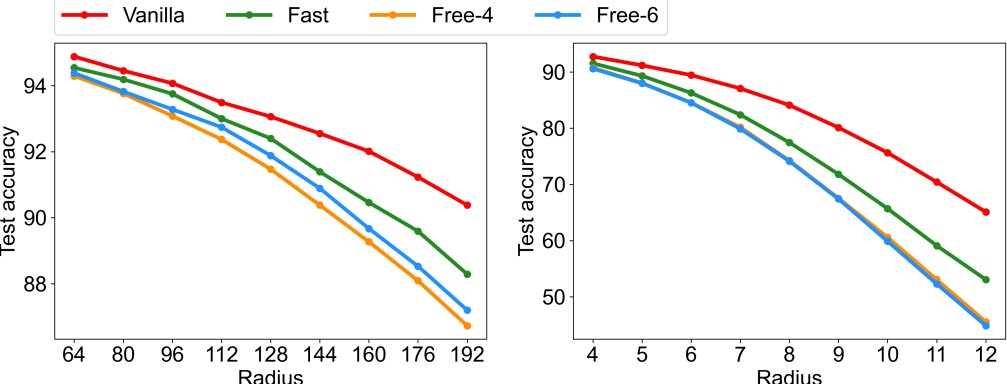

(c) Test accuracy of a standard Wide-ResNet34 target model against transferred attacks from adversarially trained ResNet18 models.

Figure 9: Test accuracy of standard ResNet18, ResNet50, and Wide-ResNet34 target models against transferred attacks from ResNet18 models adversarially trained by vanilla, fast, and free algorithms on CIFAR-10. The left figure applies $\mathcal{L}_2$ attacks of radius ranging from 64 to 192, and the right figure applies $\mathcal{L}_\infty$ attacks of radius ranging from 4 to 12.

## B.4   Generalization Gap for Different Numbers of Training Samples

We perform additional experiments to study the relationship between the number of training samples and the generalization gap. We randomly sampled a subset from the CIFAR-10 and CIFAR-100 training datasets of size $n \in \{10000, 20000, 30000, 40000, 50000\}$, and adversarially trained ResNet18 models on the subset for a fixed number of iterations. The results are demonstrated in Figure 10.

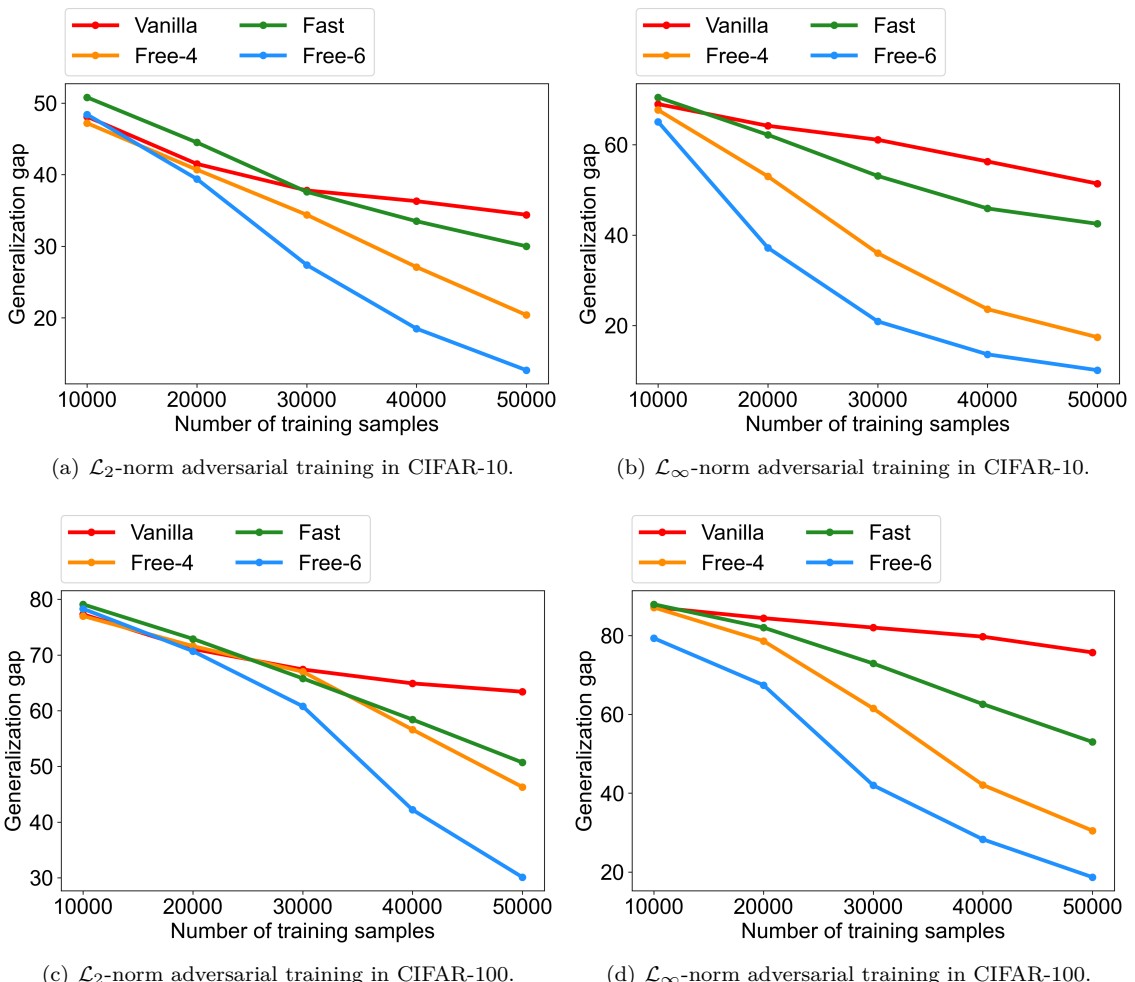

(a) $\mathcal{L}_2$-norm adversarial training in CIFAR-10.

(b) $\mathcal{L}_\infty$-norm adversarial training in CIFAR-10.

(c) $\mathcal{L}_2$-norm adversarial training in CIFAR-100.

(d) $\mathcal{L}_\infty$-norm adversarial training in CIFAR-100.

Figure 10: Adversarial generalization gap of ResNet18 models adversarially trained by vanilla, fast, and free algorithms (with free steps $m = 4$ or $m = 6$) for a fixed number of steps on a subset of CIFAR-10 or CIFAR-100.

### B.5 Hyperparameter Choices and Generalization Behavior with Early Stopping

For the fast adversarial training algorithm, we applied the learning rate of adversarial attack $\alpha_\delta = 7/255$ for $\mathcal{L}_\infty$ attack and $\alpha_\delta = 64/255$ for $\mathcal{L}_2$ attack over 200 training epochs, following from Andriushchenko & Flammarion (2020). We note that our hyperparameter selection is different from the original implementation of fast AT (Wong et al., 2020), which applied the attack step size $\alpha_\delta = 10/255$ for $\mathcal{L}_\infty$ attack over 15 training epochs with early stopping. We clarify that we do not use early stopping in our implementation to test the generalization behavior of fast AT over the same 200 number of epochs as we choose for PGD and free AT, where the choice of 200 epoch numbers follows from the reference Rice et al. (2020) studying overfitting in adversarial training. Regarding the choice of attack step size, we observed that setting the fast attack learning rate as $\alpha_\delta = 10/255$ will lead to fast AT's *catastrophic overfitting* over the course of 200 training epochs. The learning curves of fast AT with learning rate $\alpha_\delta = 7/255$ and $\alpha_\delta = 10/255$ for $\mathcal{L}_\infty$ attack over 200 training epochs are demonstrated in Figure 11. The catastrophic overfitting phenomenon in fast AT has been similarly observed and reported in the literature (Andriushchenko & Flammarion, 2020; Kim et al., 2021; Huang et al., 2023).

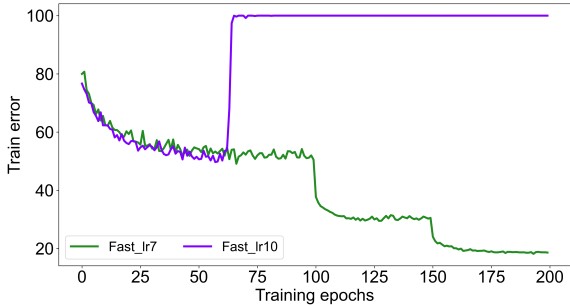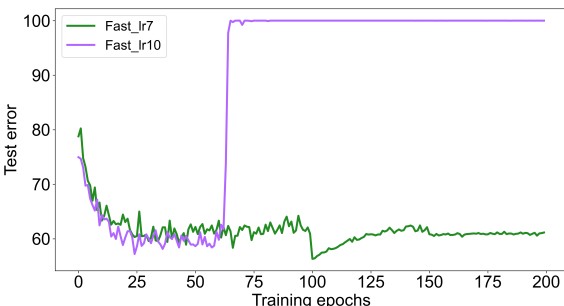

Figure 11: Learning curves of fast AT with attack learning rate $\alpha_\delta = 7/255$ and $\alpha_\delta = 10/255$ for a ResNet18 model adversarially trained against $\mathcal{L}_\infty$ attacks on CIFAR-10.

We also note that as the stability framework gives a generalization bound in terms of the min and max steps, our theoretical analysis is also applicable if early stopping is employed. From the learning curves in Figure 5, we note that throughout the whole training process, the generalization performance of free AT is consistently better than vanilla AT, which is consistent with our theoretical results. For instance, Table 6 reports the generalization gaps of the models trained by vanilla, fast (with $\alpha_\delta = 10/255$), and free AT algorithms if the training process is stopped at the 50th epoch.

Table 6: Robust training, testing, and generalization performance of the models trained by vanilla, fast (with $\alpha_\delta = 10/255$), and free AT algorithms against $\mathcal{L}_\infty$ attack on CIFAR-10, where the training process is stopped at the 50th epoch.

| Results (%) | Vanilla | Fast | Free |
|---|---|---|---|
| Train Acc. | 53.7 | 41.8 | 37.0 |
| Test Acc. | 47.1 | 39.6 | 35.4 |
| Gen. Gap | 6.6 | 2.2 | 1.6 |

**B.6 Soundness of Lower-bounded Gradient Norm Assumption in Theorem 4**

In Theorem 4 we make an assumption that the norm of gradient $\nabla_\delta h(w, \delta; x)$ is lower bounded by $1/\psi$ for some constant $\psi > 0$ with probability 1 during the training process. We note that the assumption is only required for the points within the $\epsilon$-distance from the training data. To address the soundness of this assumption, we have numerically evaluated the gradient norm over the course of free-AT training on CIFAR-10 and CIFAR-100 data, indicating that the minimum gradient norm on training data is constantly lower-bounded by $\mathcal{O}(10^{-3})$ in those experiments, i.e., $\psi = \mathcal{O}(10^3)$ is an upper bounded constant.

We trained ResNet18 networks on CIFAR-10 and CIFAR-100 datasets, applying $\mathcal{L}_2$ adversary and setting the free step $m$ as 4 and 6, and we recorded the gradient norm $\|\nabla_\delta h(w, \delta_j; x_j, y_j)\|_2$ of every sample throughout the training process. The heatmaps are plotted in Figure 12.

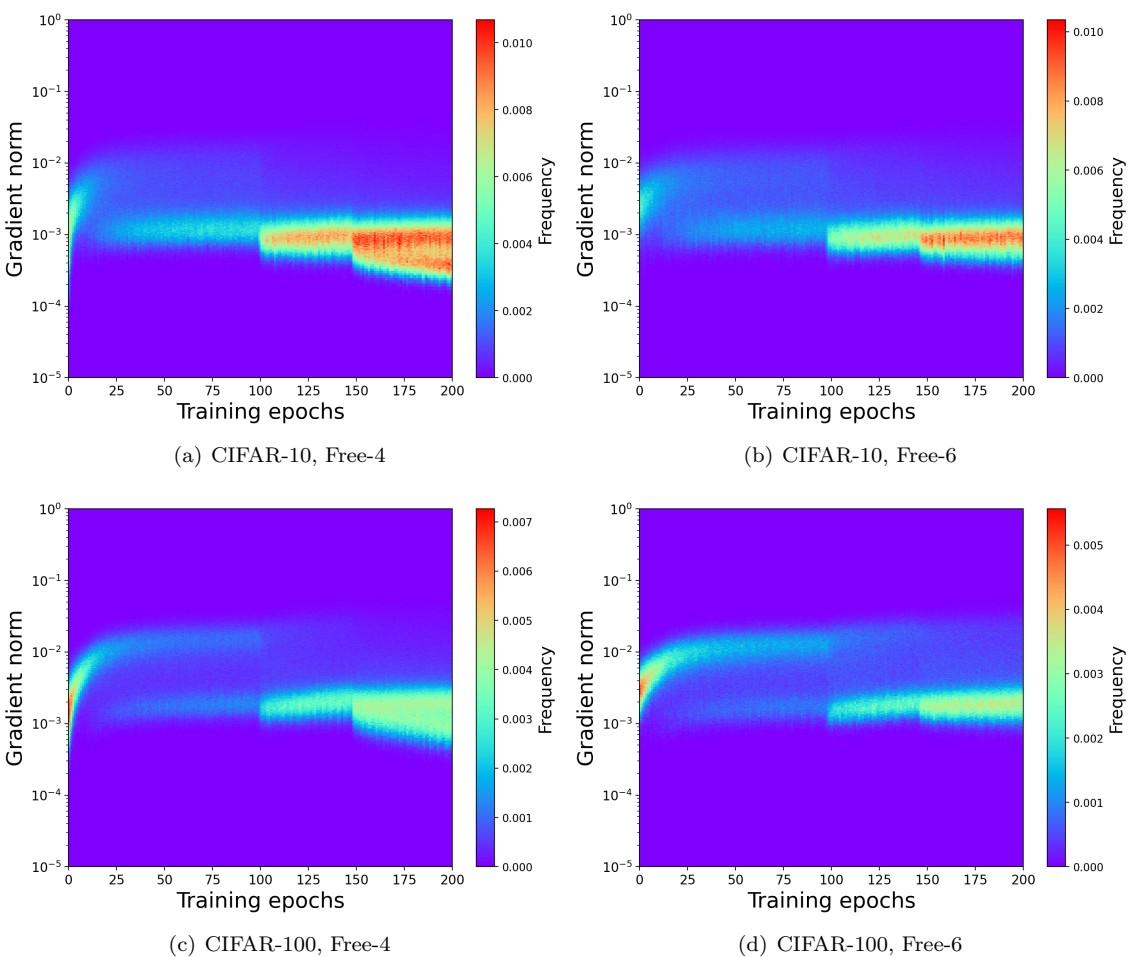

(a) CIFAR-10, Free-4

(b) CIFAR-10, Free-6

(c) CIFAR-100, Free-4

(d) CIFAR-100, Free-6

Figure 12: The heat map of gradient norm throughout the training process.

