# OpenReview forum: "Stability and Generalization in Free Adversarial Training"
_TMLR — Accepted by TMLR_

### Review · Reviewer_d8jU · 2024-09-10

**Summary Of Contributions:**

The authors highlight that traditional adversarial training, while demonstrating outstanding performance against adversarial examples, often leads to larger generalization gaps. They also observe that Free-AT, a variant of traditional adversarial training, exhibits a smaller generalization gap in comparison. \
The primary contribution of this paper is the analysis and compares the generalization behaviors across different adversarial training solutions through both theoretical tools and empirical results.
Moreover, the authors propose Free-TRADES, which combines Free-AT with the TRADES adversarial training algorithm, to further improve the generalization behavior of TRADES.

**Audience:**

Yes

**Broader Impact Concerns:**

I don’t find any ethical implications concerns should be related to this work

**Claims And Evidence:**

Yes

**Requested Changes:**

The authors analysis the bound between generalization and optimization of adversarial training w.r..t. single attack step (Fast AT and Free AI) and multi-attack step (Vanilla AT and TRADES)
However, a key analysis "adaptive step" (something between single step and multi-attack), i.e., friendly adversarial training [1] is missing.
Friendly adversarial training used curriculum training strategy, which can greatly enhance optimization, leading to better generalization.

[1] Attacks Which Do Not Kill Training Make Adversarial Learning Stronger, in ICML 2020

**Strengths And Weaknesses:**

#Strengths \
1 The authors clearly articulate the research background and motivation for their study.\
2 They support their claims with solid evidence, drawing from both theoretical and empirical results. \
3 The authors propose a promising direction for reducing the generalization gap in adversarial training (AT) development.

#Weaknesses \
1 The paper is not very well-structured, I think they misused their appendix section.
The authors claim to have made four contributions, but only three are discussed in the main text. The fourth contribution, the framework Free-TRADES, is detailed in the appendix rather than in the main body of the thesis. I believe the appendix section is for the supplementary material but not for the main contribution. I would suggest reorganizing your paper for better readability. \
2 The innovation is limited, since the main contribution mostly focused on investigating and evaluating on existing methods proposed by others rather than focusing on their own solution \
3 This paper only focuses on the DNN architecture but not consider the transformer architecture \
4 A key citation is missing (see below)

---

> ### Author Response · Authors · 2024-10-09
> **Authors' Response to Reviewer d8jU**
>
> We thank Reviewer d8jU for his/her time and constructive feedback and suggestions on our work. Below is our response to the questions and comments in the review.
>
>
> **1- Organization of the work**
>
> **Re:** As recommended by the reviewer, we moved the discussion on Free-TRADES to Section 7 in the main text, and we also transferred the theoretical analysis for fast adversarial training to Section 4 in the main text.
>
>
>
> **2- Technical contributions of this work**
>
> **Re:** While the free adversarial training method has been adopted in several applications due to its training efficiency, there is not adequate theoretical analysis toward understanding the generalization and convergence properties of this method. In this work, we attempt to show that one benefit of free AT could be its higher algorithmic stability due to its simultaneous optimization. We also note that our theoretical comparison between vanilla AT and free AT can be extended to compare TRADES and Free-TRADES.
>
> Regarding the technical contributions of our work, we note that the existing bounds in the literature do not apply to the Free-AT algorithm, because the maximization variable in Free-AT is re-initialized after the training steps are complete for every batch. One technical contribution of our work is to perform the stability analysis where the maximization variable is re-initialized after every $m$ iterations where a new mini-batch of data is used. Our theoretical results suggest that as long as $m$ is bounded by $\frac{\alpha_{\delta} \epsilon \psi}{c}$, the generalization risk will not change significantly with a greater $m$ value, which is not implied by the standard bounds in the existing works [3] in the literature. Also, our theoretical analysis considers the normalized gradient (instead of the vanilla gradient) for the gradient ascent step of solving the maximization sub-problem and mini-batch stochastic optimization for updating min and max variables at every iteration, which are not analyzed in the previous literature [3]. In the revision, we added Remark 3 to highlight our technical contributions.
>
>
>
>
> **3- Further experiments for the transformer architecture**
>
> **Re:** As recommended by the reviewer, we performed vanilla and free adversarial training algorithms on data-efficient image transformers (DeiT), following from [1]. The experimental results are presented in the following table. We similarly observed that the generalization gap of free AT is lower than vanilla AT.
>
> | CIFAR-10, $L_\infty$-norm Attack | Vanilla | Free |
> | ------ | ------ | ----- |
> | Train Accuracy (%) | 63.1 | 47.3 |
> | Test Accuracy (%) | 48.8 | 43.2 |
> | Generalization Gap (%) | 14.3 | 4.1 |
>
>
>
> **4- Analysis of Friendly adversarial training**
>
> **Re:** We thank the reviewer for pointing out the relevant reference on friendly adversarial training (Friendly-AT) [2]. In our numerical analysis, we observed that the improvement in the generalization gap by Friendly-AT depends on the training perturbation radius hyperparameter $\varepsilon_{\text{train}}$. We note that by choosing a sufficiently large $\varepsilon_{\text{train}}$ the generalization improvement over vanilla AT is similar to that of Fast-AT. On the other hand, a smaller $\varepsilon_{\text{train}}$ results in PGD AT-like numerical results, showing the trade-off in generalization-optimization accuracy explored by Friendly-AT. We have discussed the numerical observation in the revised text.
>
>
>
> | CIFAR-10, $L_2$-norm Attack | Vanilla | Friendly ($\varepsilon_{\text{train}}=128/255$) | Friendly ($\varepsilon_{\text{train}}=256/255$) |
> | ------ | ------ | ----- | ----- |
> | Train Accuracy (%) | 100.0 | 99.9 | 100.0 |
> | Test Accuracy (%) | 65.6 | 65.8 | 66.9 |
> | Generalization Gap (%) | 34.4 | 34.1 | 33.1 |
>
>
> | CIFAR-10, $L_\infty$-norm Attack | Vanilla | Friendly ($\varepsilon_{\text{train}}=8/255$) | Friendly ($\varepsilon_{\text{train}}=16/255$) |
> | ------ | ------ | ----- | ----- |
> | Train Accuracy (%) | 95.1 | 94.7 | 88.7 |
> | Test Accuracy (%) | 43.7 | 44.1 | 46.3 |
> | Generalization Gap (%) | 51.4 | 50.6 | 42.4 |
>
>
>
>
> [1] Yichuan Mo, Dongxian Wu, Yifei Wang, Yiwen Guo, Yisen Wang. When Adversarial Training Meets Vision Transformers: Recipes from Training to Architecture. Advances in Neural Information Processing Systems, 2022.
>
> [2] Jingfeng Zhang, Xilie Xu, Bo Han, Gang Niu, Lizhen Cui, Masashi Sugiyama, Mohan Kankanhalli. Attacks Which Do Not Kill Training Make Adversarial Learning Stronger. International Conference on Machine Learning, 2020
>
> [3] Farzan Farnia, and Asuman Ozdaglar. Train simultaneously, generalize better: Stability of gradient-based minimax learners. International Conference on Machine Learning, 2021.

---

### Review · Reviewer_pSfY · 2024-09-24

**Summary Of Contributions:**

This paper presents a comparative study on the generalization performance of adversarial training methods, focusing on Free Adversarial Training (Free AT) and Vanilla Adversarial Training (Vanilla AT). The authors utilize an algorithmic stability framework to derive generalization bounds for these methods. Additionally, they introduce the Free-TRADES algorithm, extend the Free AT approach to the TRADES framework, and demonstrate its advantages in terms of generalization error and robustness.

**Audience:**

Yes

**Broader Impact Concerns:**

I have no broader impact concerns.

**Claims And Evidence:**

Yes

**Requested Changes:**

While the paper is generally well-written and provides solid theoretical results regarding free adversarial training, it would benefit from clarifications on the following:

- Discussion on Robustness-Accuracy Trade-off: Elaborate on how Free AT affects the trade-off between robustness and accuracy, especially in comparison with other adversarial training methods like TRADES and Fast AT.

- Practical Impact: Discuss potential ways the theoretical bounds can guide the development of new adversarial training techniques or regularization methods.

**Strengths And Weaknesses:**

__Strengths:__

- Theoretical Novelty. The paper applies an algorithmic stability analysis to Free AT, offering new insights into its generalization properties. The generalization bounds derived for Free AT, compared to Vanilla AT, provide a theoretical foundation for its improved performance.

- Empirical Support. The empirical results validate the theoretical claims. Experiments on CIFAR-10, CIFAR-100, Tiny-ImageNet, and SVHN show that Free AT exhibits a smaller generalization gap and better robustness against adversarial attacks.

- Free-TRADES Extension. The introduction of Free-TRADES extends the applicability of Free AT to a broader class of adversarial training methods, enhancing its relevance and impact on the field.


__Weaknesses:__

- Limited Insights for Practice. The theoretical findings, while novel, do not provide strong guidance for developing new training methods or regularization techniques. The paper could benefit from a more detailed discussion of how these results can inspire practical advancements in adversarial training.

- Discussion on Robustness: The relationship between generalization gap and robustness could be elaborated further. For example, it has not been fully explored how Free AT's reduced generalization gap translates to better robustness in real-world adversarial settings. From the results from Figure 1 and Table 1, it seems that the reduced generalization gaps (w.r.t. free AT) are mainly due to the significant decrease in training robust accuracies, which requires further discussion.

---

> ### Author Response · Authors · 2024-10-09
> **Authors' Response to Reviewer pSfY**
>
> We thank Reviewer pSfY for his/her time and constructive feedback and suggestions on our work. Below is our response to the questions and comments in the review.
>
>
> **1- Practical implications**
>
> **Re:** The theoretical bounds in Theorems 2 and 4 could provide insight on the hyperparameter selection in Free-AT method. Specifically, we would like to highlight the role of the parameter $m$, which is the number of inner maximization steps, in running Free-AT for every batch of data. Our generalization bounds indicate that increasing $m$ would have a lower impact on the generalization gap compared to the number of minimization steps. This is a notable implication of the bound, as in practice, it has been observed that a larger parameter $m$ does not hurt the performance as long as it remains below 8. Theorem 4 echoes this observation as changing $m$ below the threshold $\frac{\alpha_{\delta} \epsilon \psi}{c}$ does not significantly increase the generalization error bound.
>
>
>
> **2- Relationship between generalization gap and robustness**
>
> **Re:** As pointed out by the reviewer, the growth rate of the adversarial training accuracy is more similar to that of test adversarial accuracy in Free adversarial training, that is why the generalization gap is lower in Free AT compared to Vanilla AT. We observed that the more controlled growth of the generalization gap in Free AT may lead to improvements in test accuracy against black-box adversarial attacks (Figures 2 and 3, and Appendix B.2), suggesting the impacts of controlling overfitting rates of the methods on the transferability of the robustness to non-PGD attacks. An interesting future direction could be to theoretically analyze the influence of a lower generalization gap on other performance criteria of an adversarially-trained neural net, e.g. its performance under domain shifts and black-box perturbations.
>
>
>
> **3- Robustness-accuracy trade-off**
>
> **Re:** As recommended by the reviewer, we added the discussion of the robustness-accuracy trade-off of vanilla and free adversarial training in Section 7. Theorems 2 and 4 suggest that the training process of free AT is algorithmically more stable than vanilla AT, therefore free AT could be similarly expected to generalize better on clean data than vanilla AT. The numerical results shown in the following tables are consistent with this intuition. We also observed that Free-TRADES could generalize better than TRADES on clean data.
>
>
> | CIFAR-10, $L_2$-norm Attack | Vanilla | Free | TRADES | Free-TRADES |
> | ------ | ------ | ----- |  ------ | ----- |
> | Clean Train Accuracy (%) | 100.0 | 98.7 | 99.8 | 95.8 |
> | Clean Test Accuracy (%) | 88.1 | 88.6 | 86.2 | 86.8 |
> | Clean Generalization Gap (%) | 11.9 | 10.1 | 13.6 | 9.0 |
> | Robust Train Accuracy (%) | 100.0 | 86.4 | 99.2 | 83.1 |
> | Robust Test Accuracy (%) | 65.6 | 66.0 | 66.2 | 68.0 |
> | Robust Generalization Gap (%) | 34.4 | 20.4 | 33.0 | 15.1 |
>
>
> | CIFAR-10, $L_\infty$-norm Attack | Vanilla | Free | TRADES | Free-TRADES |
> | ------ | ------ | ----- |  ------ | ----- |
> | Clean Train Accuracy (%) | 99.9 | 95.1 | 97.7 | 91.0 |
> | Clean Test Accuracy (%) | 84.7 | 85.9 | 82.4 | 83.1 |
> | Clean Generalization Gap (%) | 15.2 | 9.2 | 15.3 | 7.9 |
> | Robust Train Accuracy (%) | 95.1 | 63.7 | 85.5 | 60.2 |
> | Robust Test Accuracy (%) | 43.7 | 46.3 | 50.1 | 48.7 |
> | Robust Generalization Gap (%) | 51.4 | 17.4 | 35.4 | 11.5 |

---

### Review · Reviewer_3qGX · 2024-09-24

**Summary Of Contributions:**

This paper studies the algorithmic stability of free adversarial training and compares it with the stability of vanilla adversarial training and fast adversarial training. It considers the nonconvex-nonconcave scenario, and the results demonstrate the advantage of free adversarial training. In addition, the authors also propose a new algorithm of free-TRADES, which maintains the robust test accuracy while reduces the generalization gap.

**Audience:**

Yes

**Claims And Evidence:**

Yes

**Requested Changes:**

Please consider addressing my comments in the weaknesses section.

**Strengths And Weaknesses:**

Advantage: The writing of this paper is clear and easy to understand, and the proofs in the appendix are also clear. The proposed method demonstrates its effectiveness.

Concerns: However, there are several concerns regarding the contents of this paper:

(1) The theorems only provide the algorithmic stability of the three adversarial training methods, while the optimization convergence result is missing. From my understanding, the optimization convegence is as important as the algorithmic stability: If we always return 0 as the output, then the stability is perfect, but the optimization convergence is poor. In this situation, the training loss is large, and the testing loss will be large as well even if the gap between testing and training is zero.

(2) There is no enough highlights of the technical contributions of this paper. In particular, although the proofs of this paper is different from the common techniques used in Xing et. al. 2021 and Xiao et. al. 2022b, the steps of considering the attack update together with the parameter update is similar to handling the discriminator and the generator of GAN in

Farnia, Farzan, and Asuman Ozdaglar. "Train simultaneously, generalize better: Stability of gradient-based minimax learners." International Conference on Machine Learning. PMLR, 2021.

The authors need to highlight the key challenges when deriving the stability.

(3) There is no clear explanation or insights on why free adversarial training brings better algorithmic stability. Is the advantage of free adversarial training caused by technical tricks (because the proof of free is different from vanilla)? Or is it caused by any special property of the training algorithm?

(4) A minor issue: Theorem 5 for fast adversarial training is postponed to the appendix. Please consider move it to the main content and conduct a formal comparison with vanilla and free.

(5) What will happen if we consider convex function as in Xing et. al. 2021 and Xiao et. al. 2022b? What is the advantage of free adversarial training compared to vanilla adversarial training in this case?

---

> ### Author Response · Authors · 2024-10-09
> **Authors' Response to Reviewer 3qGX (Part 1)**
>
> We thank Reviewer 3qGX for his/her time and constructive feedback and suggestions on our work. Below is our response to the questions and comments in the review.
>
>
> **1- Optimization convergence of adversarial training methods**
>
> **Re:** We would like to clarify that the main focus of our theoretical analysis is on the generalization gap of adversarial training algorithms and studying their growth rate for the vanilla, fast, and free adversarial training methods. Our theoretical results indicate that given the same stepsize values for Free-AT and Vanilla-AT, the generalization gap of Free-AT could grow less rapidly compared to that of Vanilla-AT. However, we agree with the reviewer that studying the optimization convergence of the AT methods is also important in comparing the test performance of the methods.
>
> Please note that the Free-AT method in Algorithm 2 follows the update rule of a projected gradient descent ascent (Projected GDA) which has been widely studied in the optimization literature. To the best of our knowledge, a tight convergence rate for Projected GDA applied to a general nonconvex-nonconcave optimization problem is still an open question in the community [1,2]. As a tight convergence rate for GDA nonconvex-nonconcave optimization is not available in the literature, we relied on numerical experiments to verify the improvement in generalization gap by Free-AT method while maintaining a proper convergence rate. Note that our numerical experiments use standard selections of stepsizes and other hyperparameters for the AT algorithms, and their numerical results suggest the standard hyperparameter selection leads to a lower generalization gap for Free-AT compared to Vanilla-AT.
>
> We have added Remark 4 to the paper, explaining the existing challenges for deriving tight convergence rates for nonconvex-nonconcave optimization problems, and how we attempt to analyze the role of optimization convergence in our numerical experiments.
>
>
>
>
>
> **2- Technical contributions of this paper**
>
> **Re:** We thank the reviewer for pointing this out and we included Remark 3 in the revision to highlight our technical contributions. We note that the existing bounds in the literature do not apply to the Free-AT algorithm, because the maximization variable in Free-AT is re-initialized after the training steps are complete for every batch. One technical contribution of our work is to perform the stability analysis where the maximization variable is re-initialized after every $m$ sub-iterations where a new mini-batch of data is used. Our theoretical results suggest that as long as $m$ is bounded by $\frac{\alpha_{\delta} \epsilon \psi}{c}$, the generalization risk will not change significantly with a greater $m$ value, which is not implied by the standard bounds in the existing works [3] in the literature. Also, our theoretical analysis considers the normalized gradient (instead of the vanilla gradient) for the gradient ascent step of solving the maximization sub-problem and mini-batch stochastic optimization for updating min and max variables at every iteration, which are not analyzed in the previous literature [3].
>
>
>
> **3- Advantages of Free Adversarial Training**
>
> **Re:** Our theoretical analysis suggests that the free adversarial training method could lead to higher algorithmic stability due to its simultaneous update of the minimization and maximization variables. We note that Theorem 1 in [4] and Theorem 5.2 in [5] give a lower bound $\Omega(T/n)$ on the stability of vanilla adversarial training. On the other hand, Theorem 4 in our work suggests that the stability of free AT is upper bounded by $\mathcal{O}(T^c/n)$ for constant $c = \frac{\lambda_{\text{Free}}}{\lambda_{\text{Free}}+1} < 1$, indicating that Free-AT is more stable than Vanilla-AT. We have made this point clearer in the text.
>
>
> **4- Relocating Theorem 5 to the main content**
>
> **Re:** As recommended by the reviewer, in the revision we moved Theorem 5 for fast adversarial training to Section 5 in the main text and provided a comparison with vanilla adversarial training.

---

> ### Author Response · Authors · 2024-10-09
> **Authors' Response to Reviewer 3qGX (Part 2)**
>
> **5- Generalization analysis for convex-nonconcave problems**
>
> **Re:** For convex-nonconcave problems, Lemma 4.1 of [5] indicates that the gradient update of the model weight $w$ is bounded by $|| \nabla_w h(w_1, \delta_1; x,y) - \nabla_w h(w_2, \delta_2; x,y) || \le \beta || w_1-w_2 || + w || \delta_1-\delta_2 ||$. For vanilla adversarial training, the optimal perturbation could drastically change even when the model weight is slightly different. Therefore, the quantity $|| \delta_1-\delta_2 ||$ can only be bounded by the radius of the perturbation set, i.e., $|| \delta_1-\delta_2 || \le 2\varepsilon$. On the other hand, for free adversarial training, the perturbation is re-initialized for every batch and simultaneously updated with the model weight. Therefore, we can expect a smaller perturbation difference $||\delta_1-\delta_2||$ when the number of simultaneous optimization steps is bounded. Hence, free adversarial training is more stable than vanilla AT due to its simultaneous optimization.
>
>
>
> [1] Tianyi Lin, Chi Jin, Michael. I. Jordan. On Gradient Descent Ascent for Nonconvex-Concave Minimax Problems. International Conference on Machine Learning, 2020.
>
> [2] Haochuan Li, Farzan Farnia, Subhro Das, Ali Jadbabaie. On Convergence of Gradient Descent Ascent: A Tight Local Analysis. International Conference on Machine Learning, 2022.
>
> [3] Farzan Farnia, and Asuman Ozdaglar. Train simultaneously, generalize better: Stability of gradient-based minimax learners. International Conference on Machine Learning, 2021.
>
> [4] Yue Xing, Qifan Song, and Guang Cheng. On the algorithmic stability of adversarial training. Advances in neural information processing systems, 34:26523–26535, 2021.
>
> [5] Jiancong Xiao, Yanbo Fan, Ruoyu Sun, Jue Wang, and Zhi-Quan Luo. Stability analysis and generalization bounds of adversarial training. Advances in Neural Information Processing Systems, 35:15446–15459, 2022.

---

> > ### Comment · Reviewer_3qGX · 2024-10-21
> >
> > I appreciate the authors in addressing my concerns.

---

### Decision · Action_Editor_Ar2z · 2024-10-28

**Recommendation:** Accept with minor revision

**Comment:**

The reviewers were satisfied with the rebuttals, and as AE, I believe that this paper is technically solid enough and of interest enough to some members of the ML community to make it appropriate for publication in TMLR. The authors should be sure to provide an up-to-date, camera ready version of the paper with all of the changes fully incorporated.

**Audience:**

Yes, this paper could be used by other ML researchers, who could leverage its key insights in order to build more efficient robust learning algorithms with stronger generalization capabilities.

**Claims And Evidence:**

This paper seeks to compare the generalization gap of neural networks trained using vanilla adversarial training and free adversarial training. Using the algorithmic stability framework, the authors present proofs on the bounds on the generalization error of these methods, and conclude that free adversarial training should have a lower generalization gap. The authors support this claim with empirical evidence, showing that it holds for different models trained on a variety of datasets.

The reviewers had a variety of concerns, most notably about the specific technical contributions, their practical implications, and clarity on these issues. After rebuttal, the reviewers were satisfied with the improvements to the paper, and felt that the claims were backed by the evidence. They all recommended acceptance. Based on this, a decision to accept with minor revisions was reached.